# Realistic Evaluation of Model Merging for Compositional Generalization

**Derek Tam***  *dtam@cs.toronto.edu*
*University of Toronto*
*Vector Institute*

**Yash Kant***  *yashkant@cs.toronto.edu*
*University of Toronto*
*Vector Institute*

**Brian Lester***  *blester@cs.toronto.edu*
*University of Toronto*
*Vector Institute*

**Igor Gilitschenski**  *gilitschenski@cs.toronto.edu*
*University of Toronto*
*Vector Institute*

**Colin Raffel**  *craffel@gmail.com*
*University of Toronto*
*Vector Institute*

**Reviewed on OpenReview:** *https://openreview.net/forum?id=j7ye0nXvEm*

## Abstract

Model merging has emerged as a practical and cost-effective approach for combining multiple pretrained models into a single model that inherits their capabilities and often achieves improved performance. Its growing popularity has led to the rapid development of numerous merging techniques. However, these methods are typically evaluated in disparate experimental settings and make differing assumptions about model architecture, data availability, and computational budget, making direct comparison difficult. In this work, we systematically characterize the relative strengths and limitations of existing merging methods by evaluating them within a unified experimental framework. Our study focuses on *compositional generalization* — i.e., whether merging can successfully combine distinct skills to generalize to new settings. We also analyze the computational costs of each method and examine how performance scales as the number of merged models increases. Overall, we evaluate eight merging methods in a novel benchmark spanning three distinct cross-modal settings, resulting in 12,000 unique merge configurations. Our findings reveal the absence of a one-size-fits-all merging strategy and serves as both an outline for the holistic evaluation of future merging methods as well as a cookbook for practitioners using model merging.

## 1 Introduction

Touvron et al. (2023a;b), Stable Diffusion Rombach et al. (2022); Podell et al. (2023), and CLIP Radford et al. (2021), alongside the development of efficient methods to finetune large models Dettmers et al. (2024); Liu et al. (2022), has led to a widespread proliferation of finetuned models that are specialized to specific use

---

*Equal contributions.

cases. Thousands of these specialized models are shared in repositories like the HuggingFace Hub.[1] *Model merging* (Raffel, 2023) aims to recycle specialized models to create new improved models that generalize to new settings. Beyond retaining, or improving, performance on tasks the constituent models were trained on, merging aims to combine the capabilities of the models to enable generalization to new tasks. Model merging has exploded in popularity due to its effectiveness and extremely low cost—e.g., many of the top models on the Open LLM Leaderboard[2] were created via model merging.

Its growing popularity has recently led to the development of numerous new merging methods (Tam et al., 2023; Matena & Raffel, 2022; Ilharco et al., 2022; Yadav et al., 2023; Jin et al., 2022; Yang et al., 2024a; Yu et al., 2023; Zhao et al., 2024b; Shah et al., 2023, *inter alia*). These methods impose different requirements in terms of computational costs, data availability, and hyperparameter tuning. We find that such considerations are rarely explicitly highlighted in past work despite heavily impacting a given method's practical applicability. In addition, most past work evaluates merging methods using the merged model's multitask performance on held-in tasks (i.e., on the tasks the original constituent models were trained on). We argue that a more meaningful application of merging is *compositional generalization*, i.e., the ability to perform new tasks that require combining the capabilities of the constituent models being merged. Apart from being more challenging, we argue compositional generalization provides a more realistic motivation for merging: Prior work has shown that merging seldom improves performance on the held-in tasks (Ilharco et al., 2022; Yadav et al., 2023), and one can therefore simply use the corresponding constituent model for a held-in task to maximize performance. On the other hand, compositional generalization can unlock new capabilities and applications that the individual models lack. When deployed in the real-world, downstream use case often do not exactly match the skills a model was trained on. In this case, compositional generalization of those individual skills would allow the model to generalize and perform well in practice. For example, one can compose the instruction-following capabilities of one model with the coding expertise of another model to obtain a system that generates code from natural language prompts. Another example is composing skills for cross-embodiment transfer so that combining the policy used for self-driving cars and the policy for robotic manipulation can result in a policy so that allows the robot to navigate and perform tasks.

Thus we develop a more unified and thorough benchmark that compares these based on both held-in performance as well as the more challenging setting of "compositional generalization." To ensure our results are not modality-specific, we run our benchmark on widely used models for image classification, image generation, and natural language processing. Since merging methods can be used to combine an arbitrary number of models, we also measure how each method scales with the number of merged models. Additionally, we explicitly enumerate and evaluate merging methods based on their hyperparameter stability, computational costs, practical requirements, and whether auxiliary data is required for merging. We find that changes in these requirements cause a large variation in the best-performing merging method. Finally, to better contextualize the performance of merging methods, we compare to the often-neglected baselines of multitask training, single-model performance, and the performance of the pretrained base model.

Our experimental results provide new insights that shed light on the promises and shortcomings of existing merging methods. Overall, we find that TIES generally provides a good trade-off between performance and various practical considerations such as prerequisites, compute, and hyperparameter tuning. In addition, we find that merging outperforms multitask training on image-generation, though merging still underperforms multitask training on the very difficult problem of cross-lingual NLP. Also, we find that held-in task performance and generalization task performance are correlated in image classification but are *anticorrelated* for natural language processing. This means evaluations that consider more than held-in performance and that are done in different modalities are critical. Finally, we find that increasing the number of models being merged tends to result in *worse* multitask performance on held-in tasks but *better* compositional generalization on unseen tasks. As a whole, this paper presents a benchmark for comparing various merging methods for both held-in tasks and compositional generalization across various modalities. To encourage realistic and comprehensive evaluation of future merging methods, we will make our code publicly available [3].

---

[1] https://huggingface.co/docs/hub/en/index
[2] https://huggingface.co/spaces/HuggingFaceH4/open_llm_leaderboard
[3] https://github.com/r-three/realistic_evaluation_of_model_merging_for_compositional_generalization

## 2 Background

Model merging aims to cheaply combine models that share an architecture and an initialization (e.g. a pretrained model) in order to create an aggregate model that retains the capabilities of the individual models. We refer to the models being merged as the "constituent" models. These are typically finetuned on different datasets that cover different tasks or domains and, therefore, have complementary capabilities. We refer to the datasets that the constituent models are trained on as "held-in". The specific goals in merging can vary, but can include improving performance on a target task, creating a multitask model, retaining the capabilities of a base model, or generalizing to new tasks. For example, a popular use case of merging involves combining finetuned variants of Stable Diffusion that have been specialized to improve the quality of a particular style or object type (e.g., merging a "lego style" model with a "cute cat" model to generate cats made out of legos) Shah et al. (2023); Zhong et al. (2024); Gu et al. (2023); Yang et al. (2024b). Merging has also been widely applied to combining finetuned variants of open-source language models to improve and broaden the capabilities of the base model (Yadav et al., 2023; Yu et al., 2023).

### 2.1 Merging Methods

In this work, we assume models do not require task-specific output heads. We use $\theta_m$, $\theta_i$ (with $i \in \{1, \ldots, M\}$), and $\theta_p$ to denote the parameters of the merged model, the $M$ constituent models, and the base model used for fine-tuning, respectively. An exhaustive comparison of merging methods is beyond the scope of this work, so we choose ones that represent a diversity of approaches (i.e. that minimize interference between models or minimize the distance in activations between the models) and that represent more widely used methods and less common methods. In total, we include the following eight merging methods (with additional methods are discussed in Section 7):

**Simple Averaging** McMahan et al. (2017); Wortsman et al. (2022b); Choshen et al. (2022) uses element-wise averaging of model parameters $\theta_m = \frac{1}{M} \sum_i \theta_i$.

**SLERP** Shoemake (1985) interpolates models along the curved path connecting them (instead of along the direct line used in simple averaging). To merge more than two models, we use MLERP Kim et al. (2024), which computes a norm-preserving average.

**Task Arithmetic** Ilharco et al. (2022) constructs a "task vector" $\tau_i = \theta_i - \theta_p$ for each constituent model. The merged model is created by adding the sum of the task vectors, scaled by a hyperparameter $\lambda$, to the the pretrained model, $\theta_m = \theta_p + \lambda \sum_i \tau_i$.

**DARE** Yu et al. (2023) extends Task Arithmetic by applying dropout to the task vectors. Parameters from each task vector are randomly zeroed out with probability $p$ using mask $M_i \sim \text{Bernoulli}(p)$ and rescaled such that the expected value of the task vector is maintained. The modified task vectors $\tau_i' = \frac{(1-M_i)\tau_i}{1-p}$ are then used as in Task Arithmetic.

**TIES** Yadav et al. (2023) improves Task Arithmetic by zeroing out values in each task vector with low magnitude. An aggregate sign for parameter is chosen based on if the positive or the negative parameters have higher total magnitude. Finally, the parameters from each model that match the aggregate sign are added as in Task Arithmetic.

**Fisher Merging** Matena & Raffel (2022) merges models by maximizing the joint posterior distribution of the constituent models parameters. Posteriors are estimated via the Laplace approximation—a normal distribution with mean $\theta_i$ and the inverse of the Fisher information matrix Amari (1998) for covariance. Fisher merging uses the closed-form solution $\theta_m = \sum_i F_i \odot \theta_i / \sum_i F_i$ where $F_i$ is the diagonal Fisher of model $i$.

**RegMean** Jin et al. (2022) merges each linear layer by finding a weight matrix that minimizes the L2 distance between the activations of constituent models and the merged model. This can be cast as least squares regression between the input and output activations for each linear layer: Let $Z_i \in \mathbb{R}^{L_i \times d}$ be the collection of $L_i$ activations, each of dimensionality $d$, computed over examples from the training data of model $i$, and let $W_i$ be the parameters of some particular linear layer in that model. The closed form solution of least squares regression yields the the merged weight matrix, $W_m = \left(\sum_i \frac{1}{L_i} Z_i^\top Z_i\right)^{-1}\left(\sum_i \frac{1}{L_i}(Z_i^\top Z_i)W_i\right)$. Other parameters are merged via simple averaging. Note that only the gram matrix of the activations for each model, $Z_i^\top Z_i$, is required for the merge.

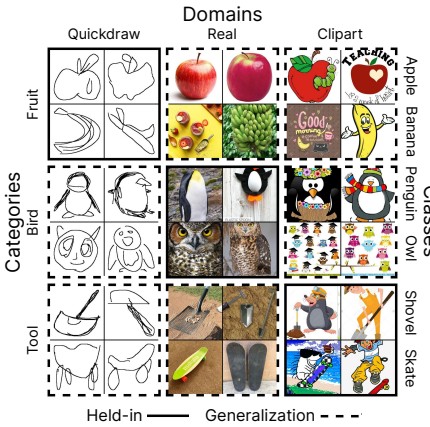

Figure 1: **Tasks used for image classification and generation.** Each row denotes objects within a category and the columns denotes domains. Each (category, domain) pair forms a task—for example, the "fruit sketch" task. Each constituent model is trained on one of the held-in tasks along the diagonal (solid border). Compositional generalization is measured via the performance on the off-diagonal tasks (dashed border).

**MaTS** Tam et al. (2023) unifies Fisher Merging and RegMean by solving a linear system that implicitly upweights models along the most important directions in parameter space. The linear system is solved using the conjugate gradient Hestenes & Stiefel (1952) method, which allows for better approximations of the Fisher Information Matrix. Parameters not in the linear layers are merged via simple averaging.

## 2.2 Challenges in Comparing Merging Methods

The rapid pace of development of merging methods has led to a lack of standardization around the experimental setup. This makes it challenging for practitioners to determine which merging method will work best for their application. In addition, past works rarely highlight the practical requirements of their novel method, such as the computational cost or availability of data. In this section, we support these assertions by clarifying the state of merging method development and evaluation.

**Different Goals** Papers presenting new merging methods often perform evaluation with different goals. For example, Matena & Raffel (2022) aim to improve performance on a "downstream" task by merging with a model trained on a "donor" task, whereas Ilharco et al. (2022) create multitask models by merging models finetuned on different tasks. In our work, we primarily focus on whether merging can enable compositional generalization by combining the capabilities of the constituent models. Evaluating compositional generalization asks: given a model finetuned to learn skills *A* and *B* and a second model that learned skills *C* and *D*, can the merged model solve a task requiring skills *A* and *C*? We argue that compositional generalization is a realistic goal as it reflects the typical use cases of merging (e.g., combining styles or objects in image generation models or enabling zero-shot generalization to new tasks for language models (Sanh et al., 2021)). Additionally, as we will demonstrate, current merging methods often struggle to provide compositional generalization, making it a challenging and meaningful evaluation setting.

**Different Experimental Setups:** Past works on merging rarely use a common experimental setup—i.e., they differ in terms of the models and datasets they consider. For example, Jin et al. (2022) validate RegMean by merging fully finetuned variants of DeBERTa-large He et al. (2020), while Yadav et al. (2023) merged variants of T5-XL-LM-Adapt Lester et al. (2021) that were finetuned using (IA)[3] Liu et al. (2022). Furthermore, Jin et al. (2022) merged models trained on 8 GLUE Wang et al. (2018) datasets whereas Yadav et al. (2023) considered 11 prompted datasets from the Public Pool of Prompts (P3) Bach et al. (2022). While sharing an experimental setup is rare, there are some exceptions—Tam et al. (2023) and Yadav et al. (2023) both replicate the setup of Ilharco et al. (2022) for their vision experiments.

**Different Prerequisites**: Simple merging methods like averaging require only the constituent model parameters to perform a merge. More sophisticated methods can require access to additional models or statistics. For example, Task Arithmetic requires access to the pretrained model that all the constituent models were finetuned from and Fisher Merging requires access to the diagonal of the Fisher Information Matrix for each. Since finetuned models are typically shared as parameter values alone, these prerequisites are often not available and/or must be separately computed. Despite the practical impact of these prerequisites, they are rarely explicitly mentioned in past work.

**Different Compute**: The computational expense of different merging methods can also vary. Many methods primarily involve element-wise operations on parameter values and therefore have relatively low computational costs. On the other hand, RegMean requires a matrix inverse for each linear layer (whose cost scales cubically with the activation dimension) and MaTS solves a linear system of equations with the conjugate gradient method. These operations incur a significant increase in computational cost, an oft-neglected consideration when new merging methods are proposed. Merging can also involve significant memory costs because naïve implementations requires loading all model parameters into memory at once. For many merging methods, it is possible to load and merge each layer individually, as done in Git-Theta Kandpal et al. (2023). Other merging methods, such as AdaMerging Yang et al. (2024a), *require* all models to be loaded simultaneously, preventing its use when memory is scarce.

**Different hyperparameter requirements**: Most merging methods have hyperparameters—for example, the scale of the task vectors in Task Arithmetic, TIES, and DARE. The sensitivity of each method to its corresponding hyperparameters is an important consideration in terms of its practical utility. In addition, tuning hyperparameters may require access to labeled validation data, which might not be available in all applications.

## 3 Comprehensive and Unified Evaluation of Merging

Given existing challenges of comparing merging methods, we propose a rigorous and comprehensive evaluation setup that aims to address these issues. In this work, we evaluate different merging methods' ability to both create a multitask model that retains performance on constituent model training tasks ("held-in") as well as generalize to new tasks or domains which are compositions of the original tasks ("generalization"). Our focus on compositional generalization is motivated by the common use of model merging to combine capabilities from different models. Concretely, consider a model fine-tuned on a task requiring skills *A* and *B*, and another model fine-tuned on a task requiring skills *A'* and *B'*. Can merging these models result in a model that can solve a task requiring skills *A* and *B'* or a task requiring skills *A'* and *B*? In the vision setting, one skill corresponds to operating across domains or styles (e.g., clipart vs. real images), while the other corresponds to image classification or generation (e.g. such as identifying the type of fruit). In the cross-lingual NLP setting, one skill corresponds to generating text in a particular language (e.g., English or Arabic), while the other corresponds to the task (e.g., summarization or question answering). We intentionally consider skills that are approximately *orthogonal*, meaning that learning one skill is not expected to directly improve performance on the other. For example, learning Korean should not inherently improve summarization ability, and learning to process clipart-style images should not improve fruit classification accuracy. By measuring compositionality of skills that are roughly "orthogonal", this ensures that any gain in performance comes from actually "composing" the "orthogonal" skills rather than a transfer of skills. Finally, we define each training task itself as a composition of underlying skills to allow for clear measurement of composition of skills. For example, though one could define skills such as arithmetic and question-answering and then measure composition of skill as a task such as question-answering over math, this would assume the composed task is fully covered and only covered by those skills. In contrast, in our setup, by defining the tasks to already be composed of underlying skills, we ensure that all required skills for the composed generalization task are already present in the individual models.

We test merging for cross-domain image classification and generation as well as cross-lingual language processing. We experiment with models that are currently widespread and have been used in past evaluations Ilharco et al. (2021); Yadav et al. (2023); Tam et al. (2023). Wherever possible, we include the performance of three oft-neglected baselines: the original pretrained model, the multitask model trained on all constituent-model datasets simultaneously, and individual-task models. For held-in tasks, the individual-task baseline is the average performance of constituent models on their respective tasks before merging. For generalization, it is the performance of models *trained* on task-specific held-out data, which are not used for training the constituent models. We do not include the baseline of evaluating constituent models on the generalization tasks since we expect poor performance due to catastrophic forgetting Vu et al. (2022).

Table 1: **(task, language) pairs used to evaluate cross-lingual compositional generalization.** Pairs marked ⭜ were used for fine-tuning; those marked ✎ were used for evaluation. Ideally, every (task, language) pair would have been used, but not all are available.

| Task ↓ / Language → | en | ar | th | de | ko |
|---|---|---|---|---|---|
| Q&A (SQuaD/XQuaD) | ⭜ | ✎ | ✎ | ✎ | |
| NLI (XNLI) | ✎ | ⭜ | ✎ | ✎ | |
| Sum. (WikiLingua) | ✎ | ✎ | ⭜ | ✎ | ✎ |
| WSD (WiC/XLWiC) | ✎ | | | ⭜ | ✎ |
| Answerability (TyDiQA) | ✎ | ✎ | ✎ | | ⭜ |

### 3.1 Cross-Domain Image Classification and Generation

For our vision experiments, we use DomainNet Peng et al. (2019), which consists of 586K images from 345 classes (e.g., "apple", "shovel", each of which has roughly 15 examples) grouped into 24 categories (e.g., "fruit", "tool") and 6 domains (e.g., "drawing", "clipart"). Fig. 1 has examples of (task, domain) combinations in DomainNet.

For each of the 24 tasks, we train a constituent model. We measure compositional generalization on the remaining domains for each category, resulting in 120 (category, domain) pairs for evaluation. See Appendix B for the full list of categories and domains. For classification, we finetune the CLIP ViT-B/32 vision encoder Radford et al. (2021), as done in previous merging work Ilharco et al. (2022); Yadav et al. (2023). To avoid task-specific classification heads, we classify by stacking CLIP's text embeddings for each label. More details are available in Appendix C.1. For generation we finetune Stable Diffusion 2.1 Rombach et al. (2022) using LoRA Hu et al. (2021) as this is currently the de facto approach to fine-tuning image generation models. More details are available in Appendix C.3.

### 3.2 Cross-Lingual Natural Language Processing

For NLP, we consider 5 distinct tasks that have datasets available in different languages. We focus on cross-lingual generalization as it is achievable via compositional generalization, but is difficult as different languages use different writing systems, vocabulary, grammatical rules, etc. Table 1 shows the chosen tasks and their available languages.

To avoid "leakage" of language-specific capabilities, we intentionally chose disparate tasks—i.e., we avoid including similar tasks such as paraphrase identification and natural language inference. Not all tasks are available in all languages, so we only evaluate cross-lingual generalization on the unseen (task, language) pairs that are available. We use mT5-xl-lm-adapt Xue et al. (2020) as our base model, a multilingual version of T5 Raffel et al. (2020) that was adapted for language modeling by Vu et al. (2022) since it is the only pretrained language model that supports all the languages we consider and comes from the same model family as previous merging papers Yadav et al. (2023). We first fully finetune mT5-xl-lm-adapt on different tasks, each in a different language. After merging the constituent models, we evaluate performance on the held-in (task, language) combinations in addition to unseen (task, language) pairs to evaluate compositional cross-lingual generalization. For all tasks, we use the standard evaluation metric and report average performance across all tasks. See Appendix C.2 for more details on training procedures and evaluation metrics.

## 4 Held-in vs. Compositional Generalization Performance

The held-in and compositional generalization performance of each merging method we consider is shown in Fig. 2. For each method, we plot the performance of the merged model on held-in datasets against the performance on datasets that require compositional generalization. For held-in performance, RegMean and MaTS work well on image classification and NLP, matching trends in previous benchmarks Tam et al. (2023), while for image generation, TIES works best. We note that in NLP, Fisher Merging tends to generalize better than RegMean and MaTS despite all three of methods implicitly minimizing the same objective Tam et al. (2023). This discrepancy could be due to Fisher Merging's looser approximation of the Fisher, which could reduce overfitting to the held-in tasks.

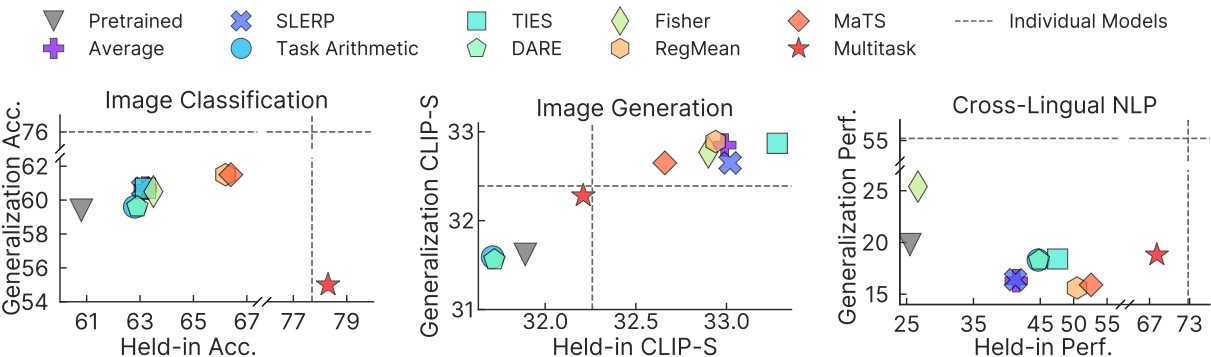

Figure 2: **Held-in vs compositional generalization performance of different merging methods in the image classification, image generation, and natural language processing.** For image classification, merging only outperforms multitask in terms of compositional generalization while for image generation it outperforms it for held-in tasks too. In NLP, compositional generalization is anticorrelated with held-in performance. Numerical values are provided in Appendix I.

For image classification, we observe a positive correlation between a merging method's held-in performance and generalization ($r$=0.828, $p$=0.011), and for image generation the correlation is stronger ($r$=0.972, $p$=5.266$e^{-5}$). This suggests that improving multitask performance can lead to improved generalization capabilities in some settings. Furthermore, for image generation, many merging methods outperform multitask training in terms of both held-in and generalization performance, highlighting the applicability and benefits of merging in this setting. In contrast, for NLP, the held-in and generalization performance is *anticorrelated* ($r$=−0.853, $p$=0.007) and most merging methods underperform the pretrained model in terms of generalization. This discrepancy could stem from cross-lingual generalization being more difficult than cross-domain generalization: we may expect vision models that can classify drawn images to be able to reasonably classify clipart images, but we would not expect a model trained on English text to be able to generate text in Arabic (which does not even share a writing system with English). Because cross-lingual generalization is a very difficult task, it is not clear if merging methods underperform multitask training due to the difficulty of cross-lingual generalization or due to some limitations of current merging methods. In addition, we may expect vision models that can classify drawn images to be able to reasonably classify clipart images, but we would not expect a model trained on English text to be able to generate text in Arabic (which does not even share a writing system with English).

In line with past work Wortsman et al. (2022b), we find that while merging tends to lag behind multitask models in terms of held-in performance, merged models can exhibit *better* generalization to new domains than the multitask and pretrained models. Taken together, our results highlight the different behaviors of merging methods are not fully captured by held-in performance, suggesting the need for evaluations that explicitly test for compositional generalization. Furthermore, the huge difference in behavior across modalities emphasizes the importance of evaluating merging methods in multiple settings.

## 5 Elucidating Practical Differences

While performance is often the main focus, we highlight that other practical requirements and costs can make different merging methods more or less attractive or applicable. We therefore argue that evaluations of merging methods should explicitly consider and explicate these considerations.

### 5.1 Prerequisites

Merging methods can differ in terms of what is required beyond the the constituent models themselves. In Table 2 we categorize these requirements for each merging method based on if it requires access to the shared **pretrained model**, requires auxiliary model **statistics** and/or requires **data access** to perform a

Table 2: **Practical differences between merging methods along different axes.** Depending on a practitioners setting, any of these differences can change which merging method is preferred. **Prerequisites: ✗** denotes merging methods that require the pretrained model parameters ($\theta_\mathbf{P}$), statistics from the pretrained model (Stat), or access to data during the merge (Data). Optional prerequisites are denoted as **✗***. **Hyperparameters (HP):** Merging methods with a ✓ require tuning their hyperparameters. The hyperparameter values we used can be found in Table 5. **Computational Cost:** Different methods also have different computational costs. "Merging" costs are incurred during each merge, while "Statistic" costs can be computed once and reused. The tables show the cost of merging a single $d{\times}k$ linear layer across $M$ models. See Appendix D for exact costs.

| | Prerequisites | | | HPs | Computational cost (FLOPs) | |
| | $\theta_\mathbf{P}$ | Stat | Data | | Merging | Statistics |
|---|---|---|---|---|---|---|
| Average | | | | | $\mathcal{O}(Mdk)$ | |
| SLERP | | | | | $\mathcal{O}(5Mdk)$ | |
| TA | ✗ | | ✗ | ✓ | $\mathcal{O}(2Mdk)$ | |
| DARE | ✗ | | ✗ | ✓ | $\mathcal{O}(6Mdk)$ | |
| TIES | ✗ | | ✗* | ✓ | $\mathcal{O}(4Mdk)$ | $\mathcal{O}(MKdk)$ |
| Fisher | | ✗* | ✗* | | $\mathcal{O}(3Mdk)$ | $4MTd^2k$ |
| RM | | ✗* | ✗* | ✓ | $\mathcal{O}(Md^2k)$ | $MTd^2k$ |
| MaTS | | ✗* | ✗* | ✓ | $\mathcal{O}((M{+}N)d^2k)$ | $4MTd^2k$ |

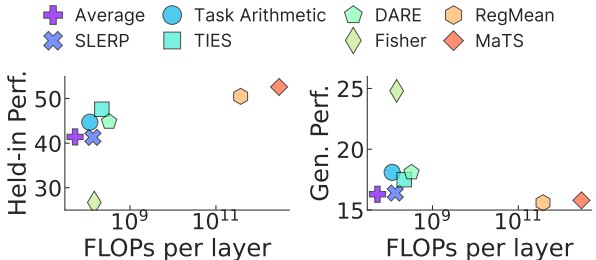

Figure 3: **The computational cost vs. performance for each merging method for NLP.** For compositional generalization, the more expensive methods are not necessarily preferred. The reported FLOPs is the upper bound for merging a single layer (see Appendix F for details and Fig. 7 (appendix) for results in all settings).

Figure 4: **Hyperparameter sensitivity of merging methods for image generation.** While TIES has the best held-in performance, it requires extensive hyperparameter tuning. We plot performance as we sweep hyperparameters. See Appendix F for a description of the hyperparameters and Fig. 8 (appendix) for results in all settings.

merge. On the whole, the merging methods we study fall into three categories: Simple averaging and SLERP have no prerequisites, which may explain their continued popularity. Task Arithmetic and its derivatives (DARE and TIES) require access to the pretrained model and data to tune hyperparameters. Finally, Fisher Merging, RegMean, and MaTS require constituent model statistics or data to compute these statistics. Such statistics are rarely shared with finetuned models which may explain why these methods are rarely used in practice despite their relatively strong performance. Methods that require access to data typically use a small validation set of ~1,000 examples. We emphasize that in applications of merging, the unavailability of a given prerequisite will make a given merging method inapplicable. Studies on merging should therefore explicitly enumerate the requirements of a given merging method.

## 5.2 Computational Costs

Another consideration when applying model merging is the amount of compute required to perform a merge as practitioners relying on model merging are more likely to lack computational resources. Table 2 shows the cost of various methods to merge a single linear layer. Broadly, we observe that some methods scale linearly while others scale quadratically in terms of $d$, the input dimension of the linear layers. As Table 2 deals with

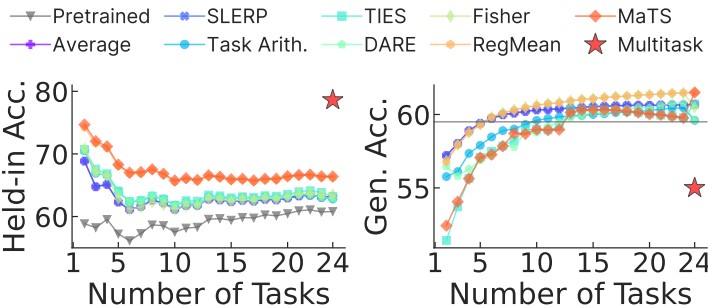

Figure 6: **Performance of merging methods as the number of constituent tasks increases for image classification.** As we merge more models, the held-in performance *decreases* while compositional generalization *increases*. We also evaluate pretrained and multitask models, on the sampled subsets. See Fig. 9 (appendix) for results in all settings.

the limiting behavior, we examine the real-world costs by plotting the number of FLOPs required for each method against performance in Fig. 3. Details of our FLOPs calculations can be found in Appendix D.

Again, we see that performance alone paints an incomplete picture of the relative merits of different merging methods. For example, in the NLP setting, more expensive methods tend to achieve better held-in performance, but the extra costs of these methods are not justifiable when considering generalization performance (with similar results for image classification and generation; see Fig. 7 for full results). It is important to note that these costs are not negligible, for example, merging 5 models using MaTS, without a hyperparameter sweep, still costs $55d^3$ FLOPs per layer while finetuning a transformer only costs $6d^2$ FLOPs. This means that with $d=4,096$, one could finetune a model on approximately 38,000 input-output pairs instead of merging. These results highlight that the lack of consideration of computational costs in prevents holistic comparison of different merging methods.

## 5.3 Hyperparameter Sensitivity

Merging methods with hyperparameters (highlighted in Table 2) often have different sensitivities to hyperparameter choice. This can heavily impact the practical utility of a given merging method. Previous works generally report performance for the best hyperparameter values, which can obscure their sensitivity. Additionally, hyperparameter tuning requires more compute and access to data, which may not always be available. We therefore measure the robustness of different merging methods to hyperparameter choice by sweeping hyperparameters as described in Appendix F using values from Table 5 for each method. The hyperparameter selection is done using the validation data on the held-in datasets.

The results of our sweep are shown in Fig. 4. We find that merging methods vary significantly in their hyperparameter sensitivity. For example, Task Arithmetic and TIES both exhibited significant sensitivity to the scaling hyperparameter $\lambda$, whereas DARE was robust to changes in the dropout probability $p$. We emphasize that differences in hyperparameter robustness can change which merging method is preferable. For example, while RegMean and MaTS tend to attain similar performance (Fig. 2), MaTS is relatively robust to hyperparameter choice (see full results in Fig. 8) and therefore may be a more attractive choice in general. Notably, trends in performance for different hyperparameter choices tend to be comparable across held-in and generalization performance, suggesting that it is relatively safe to tune hyperparameters based on held-in performance.

Hyperparameter robustness also interacts with computational costs. RegMean is already two orders of magnitudes more costly than other methods; combining this with the fact that it requires hyperparameter sweeps makes it incredibly costly overall. Conversely, TIES' high hyperparameter sensitivity necessitates tuning and therefore reduces its advantage in computational cost compared to more expensive methods like MaTS. Overall, the high variation in sensitivity to hyperparameters across merging methods highlights it as an additional important factor when comparing methods.

# 6 Scaling the Number of Models

Thus far, we have merged all possible constituent models in each experiment. In practice, the number of models being merged may vary across different applications. Therefore, we evaluate the performance of each merging method as the number of constituent models $M$ varies from 2 to 24. For each value of $M$, we draw up to 20 samples (5 for NLP and 20 for vision of $M$ constituent models, perform a hyperparameter sweep for each method, and report the average performance on the held-in and applicable generalization tasks. See Appendix G for more details. We only report the multitask performance of training on all tasks due to lack of computational resources.

Fig. 6 shows the performance as we scale the number of constituent models. Once again, we see a divergence between held-in and generalization performance; held-in performance *decreases* as the number of constituent models increases while generalization *increases*. Our held-in finding is aligned with Wortsman et al. (2022a), who found that selecting specific tasks to merged can improve held-in performance. This is likely because increasing the number of models results in increased interference, but expands the model's underlying capabilities, thus improving generalization.

We note that, with a few exceptions (e.g., Yadav et al., 2023; Ilharco et al., 2022), past studies do not measure performance as the number of constituent models change. This can cause misleading interpretations—namely, merging a large number of models and evaluating on compositional generalization may make a method appear more promising, despite this trend holding across all methods. Once again, we have shown that more comprehensive evaluation is required to provide a complete picture of merging performance.

# 7 Related Work

We focused on a subset of merging methods chosen based on their popularity and diversity. Omitted methods include: Tangent Task Arithmetic Ortiz-Jimenez et al. (2023) finetunes models in the tangent space for better weight disentanglement when using Task Arithmetic. Daheim et al. (2023) combine Fisher Merging with Task Arithmetic to help prevent the mismatch in gradients. Akiba et al. (2024) use evolutionary algorithms to choose which layers to merge. Tang et al. (2023) learn to select important parameters when merging. Jiang et al. (2023) prune task vectors for inference efficiently. Ye et al. (2023) use a gating network to predict the weights of a weighted average during inference. Tang et al. (2024) train a router between the models with unlabeled data. Other works focus on merging models that do not share initialization Ainsworth et al. (2022); Yamada et al. (2023); Singh & Jaggi (2020); Jordan et al. (2022). These are based on the hypothesis that although models lie in different loss basins Frankle et al. (2020); Juneja et al. (2022), once permutational invariances are accounted for, the models will lie in the same basin Entezari et al. (2021) and can be merged. Stoica et al. (2023) merge models with different dimensions by permuting features. Other uses of model merging include intermediate-task training Choshen et al. (2022); Gueta et al. (2023) and merging models with different modalities Sung et al. (2023).

Recent works have also measured if models can compose skills via multitask training Arora & Goyal (2023); Zhao et al. (2024a) or by combining individual-task models. For example, both Pfeiffer et al. (2020) and Vu et al. (2022) tackle cross-lingual generalization via separate "task" and "language" adapters that are swapped during inference to generalize to new (task, language) pairs. Similarly, AdaMergex uses task/language vector arithmetic for cross-lingual generalization Zhao et al. (2024b). CALM trains a cross-attention module to compose skills, but it requires access to a generalization dataset Bansal et al. (2024).

# 8 Conclusion

Our work has clarified the state of model merging by conducting an empirical study of merging methods in a comprehensive and rigorous experimental setting. The main findings of our empirical study are highlighted below:

1. Merging can outperform multitask training for certain domains such as image generation, but still underperforms multitask training in image classification and cross-lingual NLP

2. Held-in performance and generalization performance are not always correlated. For example, in cross-lingual NLP, better held-in performance is anticorrelated with better compositional generalization performance. However in other domains such as image generation, held-in performance and compositional generalization performance are correlated.

3. Compositional generalization performance increases as the number of models of merge increases though, unsurprisingly, held-in performance decreases as the number of models to merge increases. These trends plateau after roughly 10 models.

4. TIES generally provides a good trade-off between performance and various practical considerations such as prerequisites, compute, and hyperparameter tuning.

Overall, we focus on compositional generalization performance (in addition to held-in performance) and find that the two are not always correlated. Additionally, we highlight various practical factors that can render a given merging method inapplicable or unpreferable, including requirements, hyperparameter sensitivity, and computational costs. In our study, we evaluate eight merging methods in three common settings spanning natural language processing, image classification, and image generation. We hope that our released code and datasets will help accelerate the adoption of more unified evaluations of future work on model merging.

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

# A    Appendix

# B    Tasks and Domains

In DomainNet, the 24 tasks are: furniture, mammal, tool, cloth, electricity, building, office, human_body, road_transportation, food, nature, cold_blooded, music, fruit, sport, tree, bird, vegetable, shape, kitchen, water_transportation, sky_transportation, insects, and others.

The 6 domain are: clipart, infograph, painting, quickdraw, real, and sketch.

Models-to-be-merged are trained on the following (task, domain) pairs: (cloth, clipart), (furniture, clipart), (mammal, clipart), (tool, clipart), (building, infograph), (electricity, infograph), (human_body, infograph), (office, infograph), (cold_blooded, painting), (food, painting), (nature, painting), (road_transportation, painting), (fruit, quickdraw), (music, quickdraw), (sport, quickdraw), (tree, quickdraw), (bird, real), (kitchen, real), (shape, real), (vegatable, real), (insect, sketch), (others, sketch), (sky_transportation, sketch), (water_transportation, sketch).

There are 144 possible (task, domain) combinations, 24 tasks × 6 domains. Removing the 24 (task, domain) pairs used for training leaves 120 (task, domain) combinations to use for evaluation of compositional generalization.

We also note the bandage and nail classes are in both the tool task and the office task. When partitioning DomainNet into training, validation, and test sets, we follow the split used by Muqeeth et al. (2023).

Due to the paucity of data in the cross-lingual domain, all training and evaluation (task, language) pairs are enumerated in Table 1.

SQuAD Rajpurkar et al. (2016) is released under a CC BY-SA 4.0 license. WikiLingua Ladhak et al. (2020) is released under a CC0-1.0 license. XQuAD Artetxe et al. (2019) is released under a CC-BY-SA 4.0 license. XNLI Conneau et al. (2018) is released under a CC BY-NC 4.0 DEED license. WiC Pilehvar & Camacho-Collados (2018) is released under a CC BY-NC 4.0 License. XLWiC Raganato et al. (2020) is released under a CC BY-NC 4.0 License. TyDiQA Clark et al. (2020) is released under an Apache license. DomainNet Peng et al. (2019) was released under a fair use notice.

## C  Training Details and Evaluation Metrics

Models used and the training details vary based on the setting; they are outlined below. Models were trained and merged on a combination of NVIDIA A6000 and 80GB NVIDIA A100 GPUs.

### C.1  Cross-Domain Image Classification

We finetune the CLIP vision encoder from open_clip Ilharco et al. (2021); Radford et al. (2021).

Previous works finetune task-specific classifier heads on top of the CLIP representation. This means that the classifier head cannot be "merged" as different tasks have different output classes and classifier heads during evaluation. We want a classifier head that can "merged" by having all tasks use the same classifier head during evaluation. To do so, we create a frozen linear classifier head where each row vector is a representation of a class. These rows come from CLIP's textual embedding representation of the class label. Thus, loading a merged classifier head is done by embedding the text representations of each label across all tasks.

We use AdamW Loshchilov & Hutter (2017) with a learning rate $2e^{-5}$ for 1,000 steps using a batch size 128. Training is done using open_clip's $fp16$ setting. We checkpoint every 50 batches, with early stopping if validation performance does not improve after 5 checkpoints. We use accuracy as our evaluation metric.

For classification, we simply report the accuracy on held-out data. For generation, we follow standard practice and compute both the CLIP Score (CLIP-S) Hessel et al. (2021) to measure the alignment between the generated image and prompt as well as the Frechet Inception Distance (FID) Heusel et al. (2017) to measure perceptual quality. Since CLIP-S more directly measures what we aim to evaluate and we found that CLIP-S and FID were generally highly correlated in practice, we only report CLIP-S in the main text and include FID results in Appendix J.

### C.2  Cross-Lingual Language Tasks

For all tasks, we use the standard evaluation metric and report average performance across all tasks.

We finetune mT5-xl-lm-adapt Xue et al. (2020); Vu et al. (2022) using AdamW Loshchilov & Hutter (2017) with a learning rate $5e^{-4}$ for 5,000 steps using a batch size 1024. We checkpoint every 100 steps, with early stopping if validation performance does not improve after 5 checkpoints. For multiple-choice language tasks, i.e., natural language inference, word understanding, and "is question answerable", we use accuracy as the evaluation metric. For question-answering tasks, we use the average of exact-match metric Rajpurkar et al. (2016) and F1. Like Vu et al. (2022), we use SP-ROUGE to evaluate summarization tasks. SP-ROUGE is a variant of ROUGE Lin (2004) that uses language independent tokenization instead of the naïve white space character. We use the average score of SP-ROUGE variants of Rouge-1, Rouge-2, and Rouge-L.

### C.3  Cross-Domain Image Generation

We finetune Stable Diffusion 2.1 Rombach et al. (2022) using Low-Rank Adaptation (LoRA) Hu et al. (2021) on the 24 held-in (task, domain) pairs. We finetune rank 64 LoRA adapters for 10K steps using the denoising

Table 3: **Comparing compute cost of merging a linear layer between different methods.** We merge $M$ models and calculate the FLOPs required to merge a single $d{\times}k$ parameter. Two classes of methods emerge, methods that run in $\mathcal{O}(dk)$ vs. ones that run in $\mathcal{O}(d^2k)$. Precomputed statistics are calculated over $T$ tokens and often require $\mathcal{O}(MTd^2k)$ FLOPs, however, this only needs to be done once per task and can be amortized across many different merges. Note that the MLERP is the extension of SLERP used when $M{>}2$.

| Method | Merging FLOPs | Statistics FLOPs |
|---|---:|---:|
| Average | $Mdk$ | - |
| Task Arith. | $(2M+1)dk$ | - |
| DARE | $(6M+1)dk$ | - |
| TIES | $(4M+1)dk$ | $MKdk + Mdk\log(K)$ |
| Fisher | $(3M-1)dk$ | $4MTd^2k$ |
| RegMean | $(M+2)d^2k + (3M-2)dk$ | $MTd^2k$ |
| MaTS | $(M+N)d^2k + (2M+5N-2)dk$ | $4MTd^2k$ |
| SLERP | $(5M-2)dk + (M+1)\log(dk)$ | - |
| MLERP | $(2M+3)dk + (M+1)\log(dk) + \log(M)$ | - |

objective from the original work. We train with a batch size of 4 and a learning rate of $1e^{-4}$ with cosine decay using Adam Kingma & Ba (2017).

When merging models, we merge pre-multiplied A and B matrices, instead of merging the A matrices and B matrices separately, since we found this improved performance. This also requires computing model statistics are on the pre-multiplied A and B matrices.

To evaluate generated images, we use the CLIP-score Hessel et al. (2021) and FID Heusel et al. (2017) metrics. To compute held-in FID, we randomly select 3 images from each of the 345 (task, domain, class) tuples. This yields 1,035 images. Similarly, we compute generalization FID by sampling 1 image from each of the 1,722 (task, domain, class) pairs. As we have more than 1,000 images in each setting, our FID metric provides a good capture of the distribution. We use pytorch-fid Seitzer (2020) to compute FID scores with 192-dimensional features from Inception.

To select the best hyper-parameters, we use CLIP-score as an indicator for performance and sweep the same ranges described in Table 5.

## D    Computational Costs

Table 2 shows the estimated number of FLOPs required for different merging methods, as some implementations are not yet optimized. For example, MaTS uses the conjugate gradient method which requires many matrix-vector products. These are faster on GPU, but we are not aware of any linear conjugate gradient implementations on GPU, thus the time is inflated by many GPU $\leftrightarrow$ CPU transfers. However, we do include some preliminary timing results in Appendix E.

We see in Table 3 that two classes of merge methods emerge, ones that run in $\mathcal{O}(d^2k)$ and those that run in $\mathcal{O}(dk)$. Methods that run in $\mathcal{O}(d^2k)$ require a matrix multiplication while the others do not. This difference is clearer when we consider that in many transformer architectures $d = k$ and therefore these costs become $\mathcal{O}(d^3)$ and $\mathcal{O}(d^2)$.

As the majority of parameters in a transformer are from the linear layers—Attention QKV, Feed Forward layers, etc.—and some methods fallback to simple averaging for other parameters, we calculate the amount of compute required to merge a *single* linear layer. Each linear layer has an input dimension of $d$ and an output dimension of $k$ and we merge $M$ models. The conjugate gradient optimization used in MaTs is run for $N$ iterations.

When computing model statistics, we estimate the required FLOPs per token as 1 matrix multiplication in the forward pass and 3 matrix multiplications in the backward pass, following previous works which assume backward pass is 3× the forward pass Liu et al. (2022). To avoid memory issues, we pre-compute the trimming

of low magnitude parameters in TIES and only keep the top $K$ parameters. More details on this can be found in Appendix H.1. While statistic computation can be costly, it only needs to be done once per task. Thus statistics can be reused and the cost can be amortized across many different merges.

In our calculations, reduction operations across models—such as sums—require $(M-1)dk$ FLOPs and element-wise operations, such as scaling by $\lambda$, require $dk$ FLOPs. Some element-wise operations are applied to the parameter for each model independently, these require $Mdk$ FLOPs. Thus are calculations are as follows:

**Average**—$Mdk$ FLOPs. Averaging requires a sum across models and a division by the number of models.

**Task Arithmetic**—$(2M+1)dk$ FLOPs. $Mdk$ to compute the task vectors, the sum across task vectors, and two element-wise operations, scaling by $\lambda$ and adding the pretrained parameters.

**DARE**—$(6M+1)dk$ FLOPs. Assuming for simplicity that it requires 1 FLOP to generate a random number, DARE's addition of dropout requires an extra $2Mdk$ FLOPs to generate the dropout mask for each task vector—$Mdk$ FLOPs to generates the random numbers and $Mdk$ FLOPs to binarize it—$Mdk$ FLOPs to apply the masks to the task vectors, and $Mdk$ FLOPs to rescale parameters that were not dropped out. This it adds $4Mdk$ FLOPs on top of Task Arithmetic.

**TIES**—$(4M+1)dk$ FLOPs. TIES requires a sum of the trimmed parameters across models, $3dk$ to compute the sign for each parameter, find the majority sign, and replace zeros with the majority sign. $Mdk$ is required to mask each parameter, and $2(M-1)dk$ to sum the selected parameters, and the count of selected parameters, across models. The final division requires another $dk$ FLOPs.

**Fisher**—$(3M-1)dk$ FLOPs. Each model's parameters are weighted by their Fisher $Mdk$, the Fishers are summed across models as are the weighted parameters $2(M-1)dk$, and finally $dk$ FLOPs as the sum of the weighted parameters are divided by the summed Fishers.

**RegMean**—$(M+2)d^2k+(3M-2)dk$ FLOPs. The non-diagonal elements of each model's gram matrix is scaled. $Md^2k$ FLOPs are required to multiply each parameter by its respective gram matrix. These are then summed across models, as are the gram matrices. $d^2k$ FLOPs are used to invert the sum of the gram matrices and another $d^2k$ FLOPs are used to multiple the scaled parameters and the inverted sum of gram matrices.

**MaTS**—$(M+N)d^2k+(2M+5N-2)dk$ FLOPs. $Md^2k$ FLOPs are required to multiply the Fishers and the parameters for each model and $2(M-1)dk$ FLOPs are needed to sum the Fishers and scaled parameters. Each iteration of the conjugate gradient method has 1 matrix vector multiplication ($d^2k$ FLOPs), 2 inner products ($2dk$ FLOPs), and 3 vector updates $3dk$ FLOPs). If a practitioner is committed to only using MaTS merging, the Fisher-parameter multiplication can be folded into the statistics calculation and lowers the computational cost to $Nd^2k+(2M+5N-2)dk$.

**SLERP**—$(5M-2)dk+(M+1)\log(dk)$ FLOPs. $dk+\log(dk)$ FLOPs are used to calculate the norm ($dk$ for the squaring of each parameter and $\log(dk)$ for a parallelized sum of squares. The square root is constant can be ignored.). This is repeated for each of the $M$ models. Then $Mdk$ FLOPs are used to apply the calculated norms to each model. The dot product is calculated by multiplying each parameter of the two models—$2dk$, (or more generally a multiplication of the parameters from each constituent model, $(M-1)dk$—followed by a summation ($\log(dk)$). The calculations based on that dot product angle are constant, $O(1)$, with respect to the number of parameters and can be ignored. Finally $Mdk$ FLOPs are used to scale each model and $(M-1)dk$ FLOPs are used to sum the resulting models. When $M{=}2$, this cost is $8dk+3\log(dk)$ FLOPs. Some calculations, such as the models norm, require information from the whole model to be aggregated. In some implementations, these could be considered model statistics that are pre-computed and reused. This would result in a statistic cost of $Mdk+\log(dk)$ FLOPs and a merge cost of $(3M-2)dk+\log(dk)$ FLOPs.

**MLERP**—$(2M+3)dk+(M+1)\log(dk)+\log(M)$ FLOPS. Again, $Mdk+M\log(dk)$ FLOps are used to compute the norm of each model. The average model is calculated in $Mdk$ FLOPs. Then the norm of the average model is computed in $dk+\log(dk)$ FLOPs and actual normalization is applied in $dk$ FLOPs. Finally scaling by the maximum norm (found in $\log(M)$ FLOPs with a parallel implementation) is done in $dk$ FLOPs. As they are reusable across merges, the model norm calculations could be considered statistics that are pre-

computed and reused. This yields a $Mdk + M \log(dk)$ FLOPs statistic cost and a $(M+3)dk + \log(dk) + \log(M)$ merging cost.

In Fig. 7, we use the size of the transformer feed-forward layers to estimate the number of FLOPs required per layer. Feed-forward layers are generally larger than the linear layers used in attention, thus they create a upper bound on the amount of compute used to merge any linear layer. For DomainNet, we use $d=3{,}072$, $k=768$, $M=24$, and $N=50$. Similarly, we used $d=5{,}120$, $k=2{,}048$, $M=5$, and $N=50$ for the cross-lingual graphs.

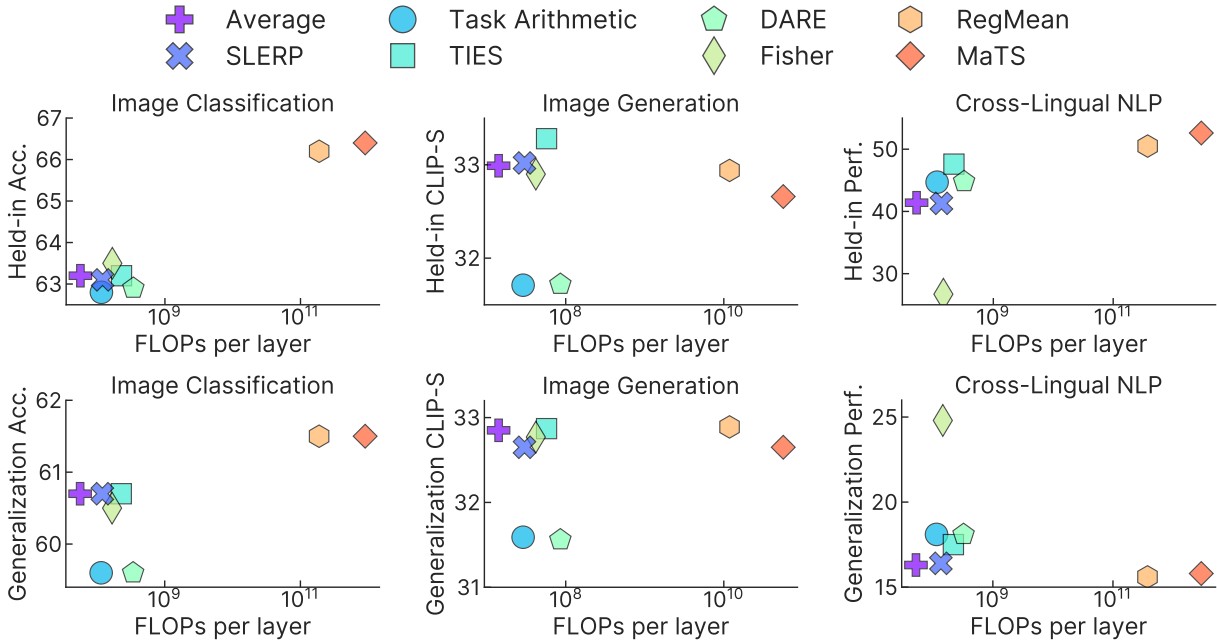

Figure 7: **The computational cost vs. performance for each merging method.** For the computational cost, we report the upper bound of the number of FLOPs required to merge a single layer (see Appendix F for details).

## E    Merging Times

Table 4 shows the amount of time required to merge a single feed-forward layer of mt5-xl-lm. We merge 5 models. The feed-forward layers are 5,120×2,048. 50 iterations of conjugate gradient were used for MaTS. We show the mean and standard deviation over 144 merges. We reiterate that the MaTS implementation is currently especially unoptimized. Despite that outlier, we see that RegMean, the only other $\mathcal{O}(d^2k)$ method, is clearly much slower than the other methods, but is still much faster than fine-tuning.

Timings were recorded on a server with 2 Intel(R) Xeon(R) Silver 4214R CPU @ 2.40GHz (12 cores/24 threads each), 256 Gigabytes of DDR4 RAM running at 2400 MT/s, and 4 NVIDIA RTX A6000 GPUs—driver version `535.129.03`—connected via PCIe 3.0×16.

## F    Hyperparameter Details

Several merging methods can be extended by including hyperparameters that scale each model-to-be-merged, i.e., a shared $\lambda$ becomes a model specific $\lambda_m$. This results in exponential growth of possible hyperparameters as more models are merged. Therefore, we do not explore per-model scaling terms; we use single, shared value of $1/M$ when an algorithm includes a scaling hyperparameter.

Similarly, some merging methods are built on top of others. For example, MaTs is initialized with Task Arithmetic, which requires running Task Arithmetic and selecting the best $\lambda$ and DARE requires two

Table 4: **Time required to merge a single feed-forward layer of mt5-xl-lm.** Timing from 144 merges with $M$=5, $d$=5,120, $k$=2,048, and $N$=50 were collected and we present the mean and standard deviation here. Again, the MaTS implementation is currently unoptimized and does many GPU to host transfers. SLERP reuslts are omitted as we no longer have the original hardware the used, making comparisons meaningless.

| Merging Method | Time (Seconds) |
|---|---|
| Average | $1.2e^{-3} \pm 000.65e^{-3}$ |
| Task Arithmetic | $1.9e^{-3} \pm 000.14e^{-3}$ |
| DARE | $2.6e^{-3} \pm 000.16e^{-3}$ |
| TIES | $2.1e^{-3} \pm 000.52e^{-3}$ |
| Fisher | $0.8e^{-3} \pm 000.27e^{-3}$ |
| RegMean | $49.9e^{-3} \pm 033.16e^{-3}$ |
| MaTS | $4,280.5e^{-3} \pm 784.38e^{-3}$ |

Table 5: **Hyperparameters considered.** We sweep hyperparameter values and select the best ones based on validation set performance. We reuse the best $\lambda$ value from Task Arithmetic for DARE.

| Method | Hyperparameters | Values |
|---|---|---|
| Average | - | - |
| SLERP | - | - |
| Task Arith. | $\lambda$: scales the task vectors | $[0.1, 1.0]$ by 0.1 |
| DARE | $\lambda$: scales the task vectors | Reused |
| | $p$: dropout probability | $[0.0, 0.9]$ by 0.1 |
| TIES | $\lambda$: scales the TIES task vectors | $[0.1, 1.0]$ by 0.1 |
| Fisher | - | - |
| RegMean | $\lambda$: scales non-diagonal elements of the gram matrices | $[0.0, 1.0]$ by 0.1 |
| MaTS | $N$: number of iterations to run conjugate gradient | $[10, 100]$ by 10 |

hyperparameters: the dropout probability $p$ and the Task Arithmetic scaling parameter $\lambda$. To reduce the space of possible hyperparameters, we first select $\lambda$—the one that works best for Task Arithmetic—and then vary the dropout probability $p$.

Fig. 8 shows the hyperparameter stability for merging methods across modalities.

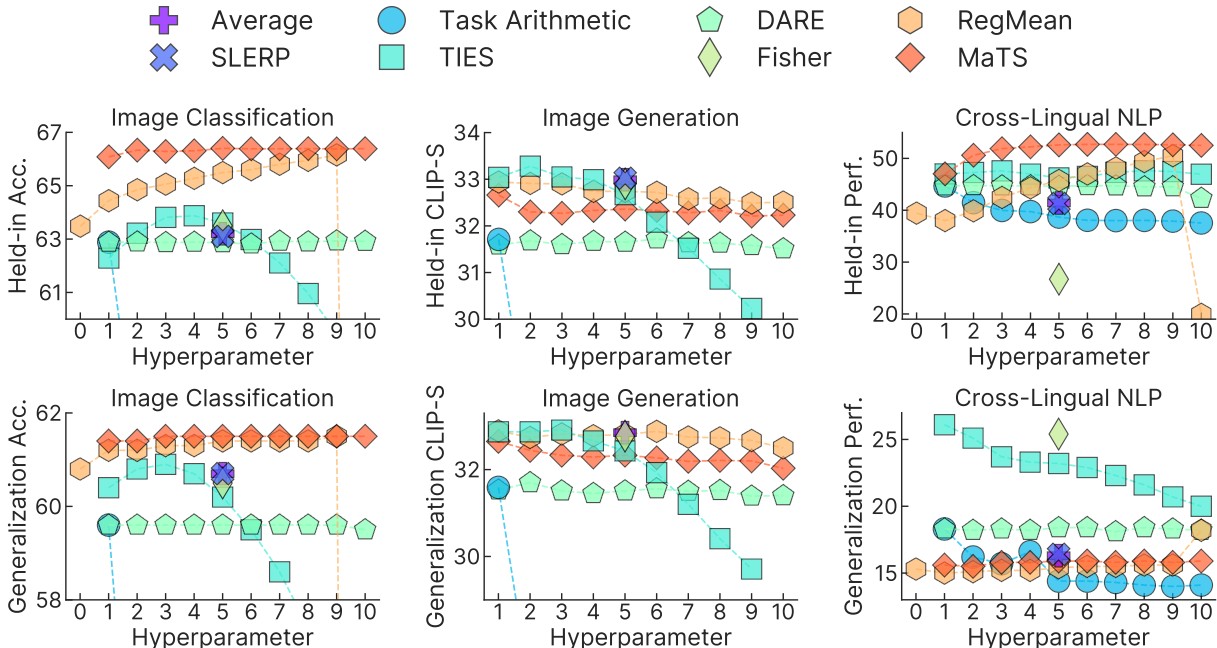

Figure 8: **Hyperparameter sensitivity of each merging method.** We plot the performance of each merging method as we sweep their respective hyperparameters. We index possible hyperparameter values from 0 to 10 as the specific hyperparameters and their ranges differ between merging methods. This captures the robustness of merging methods to different hyperparameters, regardless of the specific values. See Appendix F for a description of the hyperparameters.

## G    Sampling Procedure for Scaling the Number of Tasks

When we sample the $m$ tasks from our set of $T$ tasks to merge, we ensure that it contains all the tasks from the sample of $m-1$ tasks, i.e., the sample of $m$ tasks is the previous sample of $m-1$ tasks and a newly sampled task. For example, if the sample of 2 tasks is $\{A, C\}$, then the sample of 3 tasks will be $\{A, C, X\}$ where $X \sim T$ is a newly drawn sample. We repeat this iterative sampling procedure $L$ times and end up with $L$ different samples for each number of tasks. In the vision setup, $L$ is 20 and in the NLP setup $L$ is 5. For example, the first sample for 2 tasks might consist of $\{A, B\}$ and the first sample for 3 tasks might consists of $\{A, B, C\}$. Meanwhile, the second sample for 2 tasks might consist of $\{A, D\}$ and the first sample for 3 tasks might consists of $\{A, D, C\}$.

We use this sampling procedure to try to avoid cases where the average performance on 3 tasks is stronger than for 2 tasks simply because the 3 sampled tasks were "easier" than the 2 that were sampled.

Fig. 9 shows scaling curves across all modalities.

## H    Implementation

All of our merging methods were implemented using Git-Theta Kandpal et al. (2023) as its layer-by-layer approach allowed for merging under memory constraints.

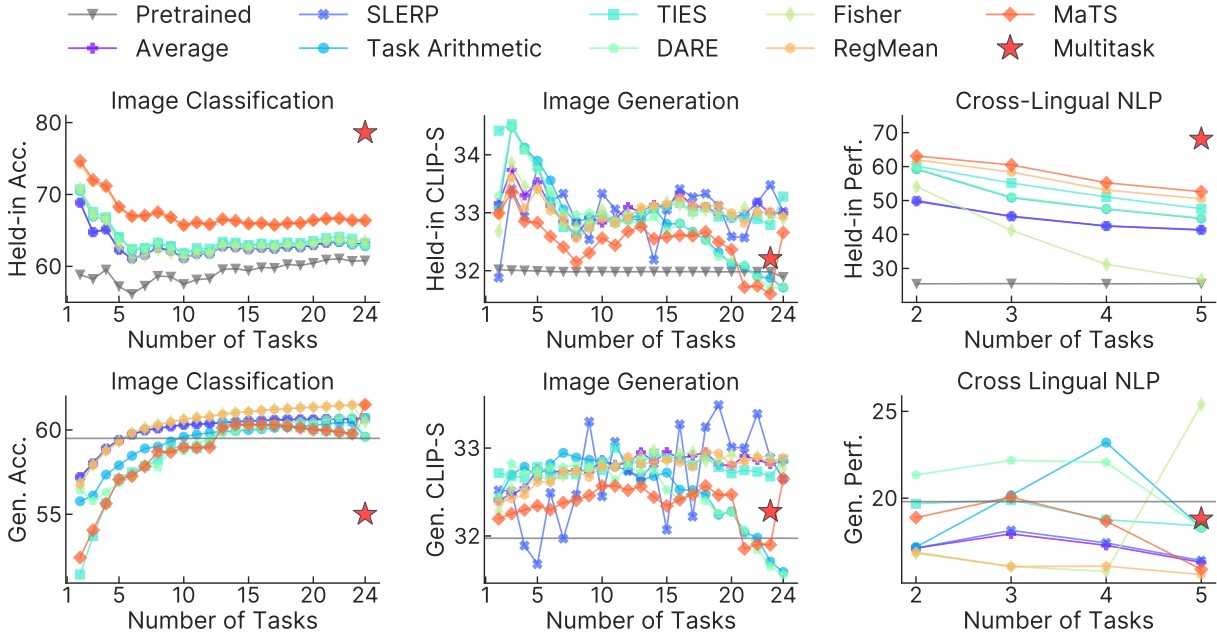

Figure 9: **Performance of merging methods as the number of constituent tasks increases.** Along the x-axis, we sample a subset of tasks 20 times for vision and 5 times for NLP and report the mean held-in and generalization performance. We additionally evaluate a pretrained model and a multitask model trained on all the held-in tasks on the sampled subsets. Since the generalization datasets and the pretrained model are fixed, its generalization performance is shown as horizontal line.

## H.1 TIES Implementation

It its original form, TIES is the only method we evaluate which does not operate on each parameter block independently. We make a few modifications to allow for parameter block independence. First, the "trim" step zeros out the parameters with the smallest magnitude across the whole model. The original implementation does this during the merge itself; however, this would require loading all of the parameters, for all constituent models, at once. To make it possible to merge 5 3.7B parameter models, we treat the "trimmed" model as a model statistic which is precomputed for each model individually, avoiding the need to load them all together. Given this statistic, ours TIES implementation merges each parameter block of all of the trimmed models independently. This is the second slight difference in our TIES implementation. In the original implementation, during the "elect" phase, parameters without an elected sign—that is, parameters whose sum across models is zero—use the majority elected sign across the *whole model*, thus ensuring that every elected sign is either positive or negative. Instead of replacing signs of zero with the majority sign across the *whole model*, we place it with the majority elected sign across the *parameter block*. The majority elected sign across the whole model cannot be pre-computed as a model statistic as it depends on all of the constituent models in the merge. It would be possible to compute the majority elected sign across the whole model by keeping a running tally as each parameter block is loaded, but it would require a second pass over the parameter blocks to apply it. Such a large change would make TIES hard to compare to other methods in terms of computational cost and time, thus we opt to make this small change in implementation to allow TIES to operate per-parameter block and make it feasible to run on our hardware.

## H.2 Fisher Merging Implementation

While the theoretical motivations of Fisher Merging require computing the Fisher on the training set, we follow Tam et al. (2023) and use the validation set since performance is comparable and it simplifies comparison to methods that require a validation set.

# I Full Results

Below we include the numerical values used in the various graphs above. Table 6, Table 7, and Table 8 are the numerical values from the left, center, and right graphs in Fig. 2.

Table 6: **Performance of different merging methods for image classification.** These are the numerical values from Fig. 2 (left).

| Merge Method | Held-In | Held-Out |
|---|---|---|
| Average | 63.2 | 60.7 |
| SLERP | 63.1 | 60.7 |
| Task Arithmetic | 62.8 | 59.6 |
| TIES | 63.2 | 60.7 |
| DARE | 62.9 | 59.6 |
| Fisher | 63.5 | 60.5 |
| RegMean | 66.2 | 61.5 |
| MaTS | 66.4 | 61.5 |
| Pretrained | 60.8 | 59.4 |
| Multitask | 78.3 | 55.0 |
| Individual Models | 77.7 | 76.0 |

Table 7: **Performance of different merging methods for image generation.** These are the numerical values from Fig. 2 (center).

| Merge Method | Held-In | Held-Out |
|---|---|---|
| Average | 32.99 | 32.85 |
| SLERP | 33.02 | 32.65 |
| Task Arithmetic | 31.71 | 31.59 |
| TIES | 33.28 | 32.87 |
| DARE | 31.72 | 31.56 |
| Fisher | 32.9 | 32.77 |
| RegMean | 32.94 | 32.89 |
| MaTS | 32.66 | 32.65 |
| Pretrained | 31.89 | 31.62 |
| Multitask | 32.21 | 32.28 |
| Individual Models | 32.26 | 32.39 |

Table 8: **Performance of different merging methods for cross-lingual NLP.** These are the numerical values from Fig. 2 (right).

| Merge Method | Held-In | Held-Out |
|---|---|---|
| Average | 41.4 | 16.3 |
| SLERP | 41.3 | 16.4 |
| Task Arithmetic | 44.7 | 18.3 |
| TIES | 47.6 | 18.4 |
| DARE | 44.8 | 18.3 |
| Fisher | 26.7 | 25.4 |
| RegMean | 50.5 | 15.6 |
| MaTS | 52.6 | 15.9 |
| Pretrained | 25.5 | 19.8 |
| Multitask | 68.1 | 18.8 |
| Individual Models | 72.8 | 55.2 |

Note that methods with no hyperparameter have their performance listed under index 5. Table 9 and Table 10 contain the numerical values used in Fig. 4 while Table 13 and Table 14 contain the values from Fig. 4.

Table 9: **Accuracy of different merging methods on the held-in tasks in the image classification setup for different hyperparameters.** These are the numerical values from Fig. 4. See Section 5.3 for a descriptions of the hyperparameters. For methods without hyperparameters, we set the hyperparameter index to 5.

| Method | Hyperparameter Index | | | | | | | | | | |
| --- | --- | --- | --- | --- | --- | --- | --- | --- | --- | --- | --- |
| | 0 | 1 | 2 | 3 | 4 | 5 | 6 | 7 | 8 | 9 | 10 |
| Pretrained | | | | | | 60.8 | | | | | |
| Average | | | | | | 63.2 | | | | | |
| SLERP | | | | | | 63.1 | | | | | |
| Task Arith. | | 62.9 | 55.0 | 36.2 | 13.6 | 2.5 | 0.4 | 0.3 | 0.3 | 0.3 | 0.3 |
| TIES | | 62.3 | 63.2 | 63.8 | 63.9 | 63.6 | 63.0 | 62.1 | 61.0 | 59.5 | 57.6 |
| DARE | 62.9 | 62.9 | 62.9 | 62.9 | 62.9 | 62.8 | 62.9 | 62.9 | 62.9 | 62.9 | |
| Fisher | | | | | | 63.5 | | | | | |
| RegMean | 63.5 | 64.4 | 64.8 | 65.1 | 65.3 | 65.5 | 65.6 | 65.8 | 66.0 | 66.2 | 0.3 |
| MaTS | | 66.1 | 66.3 | 66.3 | 66.3 | 66.4 | 66.4 | 66.4 | 66.4 | 66.4 | 66.4 |
| Multitask | | | | | | 78.6 | | | | | |

Table 10: **Accuracy of different merging methods on the generalization tasks in the image classification setup for different hyperparameters.** These are the numerical values for Fig. 4. See Section 5.3 for a descriptions of the hyperparameters. For methods without hyperparameters, we set the hyperparameter index to 5.

| Method | Hyperparameter Index | | | | | | | | | | |
| --- | --- | --- | --- | --- | --- | --- | --- | --- | --- | --- | --- |
| | 0 | 1 | 2 | 3 | 4 | 5 | 6 | 7 | 8 | 9 | 10 |
| Pretrained | | | | | | 59.4 | | | | | |
| Average | | | | | | 60.7 | | | | | |
| SLERP | | | | | | 60.7 | | | | | |
| Task Arith. | | 59.6 | 52.0 | 35.1 | 13.9 | 2.5 | 0.4 | 0.2 | 0.2 | 0.2 | 0.2 |
| TIES | | 60.4 | 60.8 | 60.9 | 60.7 | 60.2 | 59.5 | 58.6 | 57.4 | 56.0 | 54.2 |
| DARE | 59.6 | 59.6 | 59.6 | 59.6 | 59.6 | 59.6 | 59.6 | 59.6 | 59.6 | 59.5 | |
| Fisher | | | | | | 60.5 | | | | | |
| RegMean | 60.8 | 61.2 | 61.2 | 61.3 | 61.3 | 61.4 | 61.4 | 61.4 | 61.4 | 61.5 | 0.3 |
| MaTS | 61.4 | 61.4 | 61.5 | 61.5 | 61.5 | 61.5 | 61.5 | 61.5 | 61.5 | 61.5 | |
| Multitask | | | | | | 55.0 | | | | | |

Table 15, Table 17, and Table 16 include the numerical values for Fig. 3.

Table 18, Table 19, Table 20, Table 21, Table 22, and Table 23 contain the numerical values from Fig. 6.

Table 11: **CLIP score of different merging methods on the held-in tasks in the image generation setup for different hyperparameters.** These are the numerical values from Fig. 4. See Appendix F for a descriptions of the hyperparameters. For methods without hyperparameters, we set the hyperparameter index to 5.

| Method | Hyperparameter Index | | | | | | | | | | |
|---|---|---|---|---|---|---|---|---|---|---|---|
| | 0 | 1 | 2 | 3 | 4 | 5 | 6 | 7 | 8 | 9 | 10 |
| Pretrained | | | | | | 31.8 | | | | | |
| Average | | | | | | 33.0 | | | | | |
| SLERP | | | | | | 33.02 | | | | | |
| Task Arith. | | 31.7 | 27.4 | 24.8 | 23.4 | 22.5 | 22.5 | 23.5 | 23.5 | 23.3 | 23.7 |
| TIES | | 33.0 | 33.2 | 33.0 | 32.9 | 32.6 | 32.0 | 31.5 | 30.8 | 30.2 | |
| DARE | | 31.5 | 31.6 | 31.6 | 31.6 | 31.6 | 31.7 | 31.6 | 31.6 | 31.5 | 31.5 |
| Fisher | | | | | | 32.9 | | | | | |
| RegMean | | 32.9 | 32.9 | 32.9 | 32.7 | 32.7 | 32.7 | 32.5 | 32.6 | 32.4 | 32.5 |
| MaTS | | 32.6 | 32.3 | 32.2 | 32.3 | 32.3 | 32.3 | 32.2 | 32.3 | 32.2 | 32.2 |
| Multitask | | | | | | 32.2 | | | | | |

Table 12: **CLIP score of different merging methods on the generalization tasks in the image generation setup for different hyperparameters.** These are the numerical values for Fig. 4. See Appendix F for a descriptions of the hyperparameters. For methods without hyperparameters, we set the hyperparameter index to 5.

| Method | Hyperparameter Index | | | | | | | | | | |
|---|---|---|---|---|---|---|---|---|---|---|---|
| | 0 | 1 | 2 | 3 | 4 | 5 | 6 | 7 | 8 | 9 | 10 |
| Pretrained | | | | | | 31.6 | | | | | |
| Average | | | | | | 32.8 | | | | | |
| SLERP | | | | | | 32.6 | | | | | |
| Task Arith. | | 31.5 | 26.9 | 24.8 | 23.3 | 22.7 | 22.8 | 23.7 | 23.6 | 23.6 | 24.0 |
| TIES | | 32.8 | 32.8 | 32.9 | 32.6 | 32.4 | 31.9 | 31.2 | 30.4 | 29.7 | |
| DARE | | 31.5 | 31.7 | 31.5 | 31.4 | 31.5 | 31.5 | 31.5 | 31.5 | 31.3 | 31.4 |
| Fisher | | | | | | 32.7 | | | | | |
| RegMean | | 32.8 | 32.7 | 32.8 | 32.7 | 32.8 | 32.8 | 32.7 | 32.7 | 32.6 | 32.5 |
| MaTS | | 32.6 | 32.4 | 32.3 | 32.2 | 32.3 | 32.2 | 32.1 | 32.2 | 32.2 | 32.0 |
| Multitask | | | | | | 32.2 | | | | | |

Table 13: **Accuracy of different merging methods on the held-in tasks in the cross-lingual setup for different hyperparameters.** These are the numerical values from Fig. 4 See Appendix F for a descriptions of the hyperparameters.

| Method | Hyperparameter Index | | | | | | | | | | |
|---|---|---|---|---|---|---|---|---|---|---|---|
| | 0 | 1 | 2 | 3 | 4 | 5 | 6 | 7 | 8 | 9 | 10 |
| Pretrained | | | | | | 25.5 | | | | | |
| Average | | | | | | 41.4 | | | | | |
| SLERP | | | | | | 41.3 | | | | | |
| Task Arith. | | 44.7 | 41.5 | 40.1 | 39.8 | 38.7 | 38.1 | 38.0 | 38.0 | 38.0 | 37.5 |
| TIES | | 47.0 | 47.3 | 47.6 | 47.0 | 46.2 | 46.5 | 47.3 | 47.3 | 47.6 | 50.0 |
| DARE | 44.7 | 44.8 | 44.7 | 44.7 | 44.6 | 44.7 | 44.7 | 44.5 | 44.5 | 42.3 | |
| Fisher | | | | | | 26.7 | | | | | |
| RegMean | 39.4 | 38.0 | 39.9 | 42.5 | 44.3 | 45.7 | 47.0 | 48.1 | 49.4 | 50.5 | 19.2 |
| MaTS | 47.0 | 50.6 | 51.9 | 52.2 | 52.6 | 52.7 | 52.7 | 52.7 | 51.6 | 52.5 | |
| Multitask | | | | | | 68.1 | | | | | |

Table 14: **Accuracy of different merging methods on the generalization tasks in the cross-lingual setup for different hyperparameters.** These are the numerical values from Fig. 4. See Appendix F for a descriptions of the hyperparameters. For methods without hyperparameters, we set the hyperparameter index to 5.

| Method | Hyperparameter Index | | | | | | | | | | |
|---|---|---|---|---|---|---|---|---|---|---|---|
| | 0 | 1 | 2 | 3 | 4 | 5 | 6 | 7 | 8 | 9 | 10 |
| Pretrained | | | | | | 19.8 | | | | | |
| Average | | | | | | 16.3 | | | | | |
| SLERP | | | | | | 16.4 | | | | | |
| Task Arith. | | 18.3 | 16.2 | 15.7 | 16.6 | 14.4 | 14.4 | 14.3 | 14.1 | 14.0 | 14.1 |
| TIES | | 26.1 | 25.1 | 23.7 | 23.3 | 23.2 | 22.9 | 22.3 | 21.6 | 20.7 | 20.0 |
| DARE | 18.3 | 18.2 | 18.3 | 18.2 | 18.4 | 18.4 | 18.1 | 18.4 | 18.3 | 18.2 | |
| Fisher | | | | | | 25.4 | | | | | |
| RegMean | 15.3 | 15.0 | 15.1 | 15.2 | 15.2 | 15.4 | 15.5 | 15.5 | 15.6 | 15.6 | 18.2 |
| MaTS | 15.6 | 15.5 | 15.8 | 15.8 | 15.9 | 15.9 | 15.8 | 15.0 | 15.8 | 15.9 | |
| Multitask | | | | | | 18.8 | | | | | |

Table 15: **Computational Cost and Performance for image classification on DomainNet.** These are the numerical values for Fig. 3.

| Method | Compute Cost | Held-in Acc. | Generalization Acc. |
|---|---|---|---|
| Average | 1,843,200 | 63.2 | 60.7 |
| SLERP | 3,917,084 | 63.1 | 60.7 |
| Task Arith. | 3,763,200 | 62.8 | 59.6 |
| TIES | 7,449,600 | 63.2 | 60.7 |
| DARE | 11,136,000 | 62.9 | 59.6 |
| Fisher | 5,452,800 | 63.5 | 60.5 |
| RegMean | 1,538,918,400 | 66.2 | 61.5 |
| MaTS | 183,792,691,200 | 66.4 | 61.5 |

Table 16: **Computational Cost and Performance for DomainNet generation.** These are the numerical values for Fig. 3.

| Method | Compute Cost | Held-in CLIP-S | Generalization CLIP-S |
|---|---|---|---|
| Average | 1,843,200 | 32.99 | 32.85 |
| SLERP | 3,917,084 | 33.02 | 32.65 |
| Task Arith. | 3,763,200 | 31.71 | 31.59 |
| TIES | 7,449,600 | 33.28 | 32.87 |
| DARE | 11,136,000 | 31.72 | 31.56 |
| Fisher | 5,452,800 | 32.9 | 32.77 |
| RegMean | 1,538,918,400 | 32.94 | 32.89 |
| MaTS | 47,012,505,600 | 32.66 | 32.65 |

Table 17: **Computational Cost and Performance in the cross-lingual setting.** These are the numerical values for Fig. 3.

| Method | Compute Cost | Held-in Perf. | Generalization Perf. |
|---|---|---|---|
| Average | 512,000 | 41.4 | 16.3 |
| SLERP | 1,331,270 | 41.3 | 16.4 |
| Task Arith. | 1,126,400 | 44.7 | 18.1 |
| TIES | 2,150,400 | 47.6 | 17.5 |
| DARE | 3,174,400 | 44.8 | 18.1 |
| Fisher | 1,433,600 | 26.7 | 24.8 |
| RegMean | 1,469,337,600 | 50.5 | 15.6 |
| MaTS | 1,077,412,659,200 | 52.6 | 15.8 |

Table 18: **Average accuracy (across 10 different samples) of different merging methods on the held-in tasks in the image classification setup when merging various number of tasks (#T).** These are the numerical values from Fig. 6. The multitask performance and pretrained model performance can be found in Table 10. TA stands for Task Arithmetic and RM for RegMean.

| #T | Merge Method | | | | | | | | |
|---|---|---|---|---|---|---|---|---|---|
| | Pre. | Avg. | SLERP | TA | TIES | DARE | Fisher | RM | MaTS |
| 2 | 58.9 | 68.9 | 68.9 | 70.5 | 70.7 | 70.7 | 70.8 | 74.2 | 74.7 |
| 3 | 58.2 | 64.8 | 64.8 | 66.9 | 67.6 | 67.0 | 67.2 | 71.7 | 72.0 |
| 4 | 59.5 | 65.1 | 65.1 | 66.6 | 66.8 | 66.6 | 66.6 | 70.9 | 71.2 |
| 5 | 57.2 | 62.3 | 62.3 | 63.9 | 64.1 | 63.9 | 62.9 | 68.1 | 68.3 |
| 6 | 56.1 | 61.1 | 61.1 | 62.4 | 62.4 | 62.5 | 61.2 | 66.8 | 67.0 |
| 7 | 57.3 | 61.6 | 61.6 | 62.5 | 62.5 | 62.6 | 61.5 | 66.8 | 67.1 |
| 8 | 58.7 | 62.6 | 62.6 | 63.2 | 63.3 | 63.4 | 62.4 | 67.4 | 67.5 |
| 9 | 58.5 | 62.2 | 62.2 | 62.8 | 62.8 | 62.7 | 61.9 | 66.7 | 66.8 |
| 10 | 57.5 | 61.1 | 61.2 | 61.8 | 61.8 | 61.8 | 61.1 | 65.5 | 65.8 |
| 11 | 58.2 | 61.8 | 61.8 | 62.3 | 62.5 | 62.3 | 61.7 | 66.0 | 66.1 |
| 12 | 58.3 | 61.8 | 61.8 | 62.0 | 62.5 | 62.0 | 61.8 | 65.7 | 65.9 |
| 13 | 59.6 | 62.8 | 62.8 | 63.1 | 63.4 | 63.1 | 62.9 | 66.4 | 66.6 |
| 14 | 59.7 | 62.7 | 62.7 | 63.1 | 63.3 | 63.1 | 62.7 | 66.2 | 66.4 |
| 15 | 59.4 | 62.4 | 62.4 | 62.8 | 63.0 | 62.8 | 62.4 | 65.8 | 66.0 |
| 16 | 59.8 | 62.6 | 62.6 | 63.1 | 63.2 | 63.1 | 62.7 | 65.9 | 66.0 |
| 17 | 59.8 | 62.5 | 62.5 | 62.9 | 63.0 | 62.9 | 62.6 | 65.6 | 65.8 |
| 18 | 60.3 | 62.8 | 62.8 | 63.2 | 63.3 | 63.2 | 63.0 | 65.9 | 66.0 |
| 19 | 60.1 | 62.7 | 62.7 | 63.0 | 63.2 | 63.0 | 62.8 | 65.9 | 66.0 |
| 20 | 60.5 | 63.0 | 63.0 | 63.2 | 63.6 | 63.3 | 63.2 | 66.3 | 66.4 |
| 21 | 60.9 | 63.3 | 63.3 | 63.5 | 64.0 | 63.5 | 63.6 | 66.5 | 66.6 |
| 22 | 61.1 | 63.4 | 63.4 | 63.5 | 64.1 | 63.5 | 63.8 | 66.5 | 66.7 |
| 23 | 60.7 | 63.1 | 63.1 | 63.0 | 63.9 | 63.0 | 63.5 | 66.2 | 66.4 |

Table 19: **Average accuracy (across 10 different samples) of different merging methods on the generalization tasks in the image classification setup when merging various number of tasks (#T).** These are the numerical values for Fig. 6. The multitask performance and pretrained model performance can be found in Table 9. TA stands for Task Arithmetic and RM for RegMean.

| #T | Merge Method | | | | | | | |
|---|---|---|---|---|---|---|---|---|
| | **Avg.** | **SLERP** | **TA** | **TIES** | **DARE** | **Fisher** | **RM** | **MaTS** |
| 2 | 57.2 | 57.2 | 55.8 | 51.4 | 56.4 | 56.8 | 56.8 | 52.4 |
| 3 | 58.0 | 58.0 | 56.1 | 53.7 | 55.8 | 57.9 | 57.9 | 54.1 |
| 4 | 58.9 | 58.9 | 57.4 | 55.6 | 56.3 | 58.8 | 58.8 | 55.6 |
| 5 | 59.4 | 59.4 | 57.9 | 57.1 | 56.9 | 59.3 | 59.3 | 57.1 |
| 6 | 59.7 | 59.7 | 58.5 | 57.5 | 57.2 | 59.8 | 59.8 | 57.3 |
| 7 | 60.0 | 60.0 | 58.9 | 57.9 | 57.9 | 60.1 | 60.1 | 57.8 |
| 8 | 60.1 | 60.1 | 59.0 | 58.2 | 57.8 | 60.3 | 60.3 | 58.7 |
| 9 | 60.2 | 60.2 | 59.3 | 59.0 | 58.8 | 60.5 | 60.5 | 58.7 |
| 10 | 60.3 | 60.3 | 59.6 | 59.0 | 58.8 | 60.6 | 60.6 | 59.0 |
| 11 | 60.3 | 60.3 | 59.7 | 58.9 | 59.2 | 60.7 | 60.7 | 58.9 |
| 12 | 60.4 | 60.4 | 59.8 | 59.1 | 59.5 | 60.8 | 60.8 | 59.0 |
| 13 | 60.4 | 60.4 | 59.9 | 60.3 | 59.8 | 61.0 | 60.9 | 60.1 |
| 14 | 60.5 | 60.5 | 59.9 | 60.3 | 59.9 | 61.0 | 61.0 | 60.3 |
| 15 | 60.5 | 60.5 | 60.0 | 60.3 | 60.2 | 61.1 | 61.1 | 60.3 |
| 16 | 60.5 | 60.6 | 60.0 | 60.3 | 60.1 | 61.1 | 61.1 | 60.3 |
| 17 | 60.6 | 60.6 | 60.1 | 60.3 | 60.3 | 61.2 | 61.2 | 60.3 |
| 18 | 60.6 | 60.6 | 60.2 | 60.2 | 60.4 | 61.2 | 61.3 | 60.2 |
| 19 | 60.6 | 60.6 | 60.2 | 60.1 | 60.5 | 61.3 | 61.3 | 60.1 |
| 20 | 60.6 | 60.6 | 60.3 | 60.0 | 60.5 | 61.3 | 61.4 | 60.0 |
| 21 | 60.6 | 60.6 | 60.4 | 60.0 | 60.6 | 61.4 | 61.4 | 60.0 |
| 22 | 60.6 | 60.6 | 60.4 | 59.9 | 60.7 | 61.4 | 61.5 | 59.9 |
| 23 | 60.6 | 60.6 | 60.5 | 59.8 | 60.8 | 61.5 | 61.5 | 59.8 |

Table 20: **CLIP score of different merging methods on the held-in tasks in the image generation setup when merging various number of tasks (#T).** These are the numerical values from Fig. 6. The multitask performance and pretrained model performance can be found in Table 11. TA stands for Task Arithmetic and RM for RegMean.

| #T | Merge Method | | | | | | | |
|---|---|---|---|---|---|---|---|---|
| | Avg. | SLERP | TA | TIES | DARE | Fisher | RM | MaTS |
| 2 | 33.1 | 31.8 | 33.0 | 34.4 | 33.2 | 32.6 | 32.9 | 32.9 |
| 3 | 33.7 | 33.4 | 34.4 | 34.5 | 34.4 | 33.8 | 33.6 | 33.3 |
| 4 | 33.3 | 32.9 | 34.1 | 34.0 | 34.0 | 33.4 | 33.0 | 32.8 |
| 5 | 33.5 | 33.7 | 33.8 | 33.8 | 33.7 | 33.4 | 33.4 | 32.8 |
| 6 | 33.1 | 33.1 | 33.5 | 33.2 | 33.3 | 33.1 | 33.0 | 32.5 |
| 7 | 32.8 | 33.3 | 33.0 | 32.7 | 32.9 | 32.8 | 32.9 | 32.4 |
| 8 | 32.6 | 32.8 | 32.8 | 32.6 | 32.9 | 32.7 | 32.5 | 32.1 |
| 9 | 32.9 | 32.5 | 32.9 | 32.7 | 32.9 | 32.8 | 32.7 | 32.3 |
| 10 | 32.8 | 33.3 | 32.9 | 32.9 | 32.9 | 33.0 | 32.8 | 32.5 |
| 11 | 32.8 | 33.0 | 32.8 | 32.8 | 32.8 | 32.8 | 32.8 | 32.4 |
| 12 | 33.0 | 32.8 | 32.8 | 32.9 | 32.8 | 32.9 | 32.9 | 32.6 |
| 13 | 33.0 | 32.9 | 32.8 | 32.9 | 32.8 | 33.0 | 33.0 | 32.7 |
| 14 | 33.1 | 32.1 | 33.0 | 33.0 | 32.9 | 33.0 | 33.1 | 32.5 |
| 15 | 33.0 | 33.0 | 32.8 | 33.1 | 32.6 | 33.1 | 33.1 | 32.5 |
| 16 | 33.3 | 33.4 | 32.8 | 33.1 | 32.8 | 33.1 | 33.2 | 32.6 |
| 17 | 33.1 | 33.2 | 32.6 | 33.0 | 32.6 | 32.9 | 33.1 | 32.6 |
| 18 | 33.0 | 33.3 | 32.5 | 33.0 | 32.5 | 33.0 | 33.1 | 32.6 |
| 19 | 32.9 | 33.1 | 32.3 | 32.9 | 32.2 | 32.9 | 33.1 | 32.4 |
| 20 | 32.9 | 32.5 | 32.1 | 32.9 | 32.1 | 32.9 | 33.0 | 32.3 |
| 21 | 32.9 | 32.5 | 32.0 | 32.9 | 32.1 | 33.0 | 32.8 | 31.7 |
| 22 | 33.1 | 33.1 | 31.9 | 32.8 | 31.8 | 33.0 | 33.0 | 31.7 |
| 23 | 33.0 | 33.5 | 31.9 | 32.8 | 31.7 | 32.9 | 33.0 | 31.6 |
| 24 | 33.0 | 33.0 | 31.7 | 33.3 | 31.7 | 32.9 | 32.9 | 32.7 |

Table 21: **CLIP score of different merging methods on the generalization tasks in the image generation setup when merging various number of tasks (#T).** These are the numerical values from Fig. 6. The multitask performance and pretrained model performance can be found in Table 12. TA stands for Task Arithmetic and RM for RegMean.

| #T | Merge Method | | | | | | | |
|---|---|---|---|---|---|---|---|---|
| | Avg. | SLERP | TA | TIES | DARE | Fisher | RM | MaTS |
| 2 | 32.4 | 32.5 | 32.4 | 32.7 | 32.4 | 32.2 | 32.3 | 32.1 |
| 3 | 32.4 | 32.5 | 32.6 | 32.6 | 32.8 | 32.5 | 32.4 | 32.2 |
| 4 | 32.5 | 31.8 | 32.7 | 32.6 | 32.7 | 32.5 | 32.4 | 32.2 |
| 5 | 32.7 | 31.6 | 32.6 | 32.7 | 32.7 | 32.6 | 32.6 | 32.3 |
| 6 | 32.7 | 32.4 | 32.8 | 32.7 | 32.7 | 32.6 | 32.6 | 32.2 |
| 7 | 32.7 | 31.9 | 32.9 | 32.7 | 32.7 | 32.6 | 32.7 | 32.3 |
| 8 | 32.7 | 32.4 | 32.8 | 32.7 | 32.8 | 32.7 | 32.7 | 32.4 |
| 9 | 32.7 | 33.3 | 32.8 | 32.7 | 32.7 | 32.7 | 32.6 | 32.4 |
| 10 | 32.8 | 32.4 | 32.8 | 32.7 | 32.8 | 32.7 | 32.7 | 32.5 |
| 11 | 32.8 | 33.0 | 32.7 | 33.0 | 32.6 | 32.8 | 32.7 | 32.5 |
| 12 | 32.8 | 32.7 | 32.7 | 32.8 | 32.7 | 32.8 | 32.9 | 32.5 |
| 13 | 32.9 | 32.6 | 32.6 | 32.8 | 32.7 | 32.9 | 32.8 | 32.5 |
| 14 | 32.8 | 32.8 | 32.6 | 32.9 | 32.7 | 32.9 | 32.8 | 32.4 |
| 15 | 32.9 | 32.0 | 32.7 | 32.8 | 32.5 | 32.8 | 32.8 | 32.3 |
| 16 | 32.8 | 33.2 | 32.5 | 32.8 | 32.4 | 32.9 | 32.8 | 32.4 |
| 17 | 32.9 | 32.2 | 32.5 | 32.8 | 32.4 | 32.9 | 32.7 | 32.4 |
| 18 | 32.9 | 33.2 | 32.4 | 32.9 | 32.3 | 32.8 | 32.9 | 32.5 |
| 19 | 32.8 | 33.4 | 32.2 | 32.7 | 32.2 | 32.8 | 32.9 | 32.4 |
| 20 | 32.7 | 33.0 | 32.2 | 32.7 | 32.2 | 32.9 | 32.8 | 32.4 |
| 21 | 32.9 | 33.0 | 32.0 | 32.7 | 32.0 | 32.8 | 32.8 | 31.8 |
| 22 | 32.8 | 33.3 | 31.9 | 32.7 | 31.8 | 32.9 | 32.8 | 31.9 |
| 23 | 32.8 | 32.9 | 31.7 | 32.7 | 31.7 | 32.9 | 32.8 | 31.9 |
| 24 | 32.9 | 32.7 | 31.6 | 32.9 | 31.6 | 32.8 | 32.9 | 32.7 |

Table 22: **Average performance (across 5 different samples) of different merging methods on the held-in tasks in the cross-lingual setup when merging various number of tasks (#T).** These are the numerical values from Fig. 6. The multitask performance and pretrained model performance can be found in Table 13. TA stands for Task Arithmetic and RM for RegMean.

| #T | | Merge Method | | | | | | |
|---|---|---|---|---|---|---|---|---|
| | Pre. | Avg. | SLERP | TA | TIES | DARE | Fisher | RM | MaTS |
| 2 | 25.5 | 49.8 | 50.0 | 59.2 | 60.1 | 59.5 | 54.1 | 62.0 | 63.1 |
| 3 | 25.5 | 45.4 | 45.3 | 50.9 | 55.1 | 51.0 | 41.2 | 58.4 | 60.5 |
| 4 | 25.5 | 42.5 | 42.5 | 47.5 | 51.0 | 47.6 | 31.2 | 53.1 | 55.2 |

Table 23: **Average performance (across 5 different samples) of different merging methods on the generalization tasks in the cross-lingual setup when merging various number of tasks (#T).** These are the numerical values for Fig. 6. The multitask performance and pretrained model performance can be found in Table 14. TA stands for Task Arithmetic and RM for RegMean.

| #T | Merge Method | | | | | | | |
|---|---|---|---|---|---|---|---|---|
| | Avg. | SLERP | TA | TIES | DARE | Fisher | RM | MaTS |
| 2 | 17.1 | 17.1 | 17.2 | 19.7 | 21.3 | 16.8 | 16.9 | 18.9 |
| 3 | 17.9 | 18.1 | 20.2 | 19.9 | 22.2 | 16.1 | 16.1 | 20.1 |
| 4 | 17.3 | 17.4 | 23.2 | 18.8 | 22.1 | 15.8 | 16.1 | 18.7 |

## J   FID results from Image Generation on DomainNet

Along with CLIP-score, we also evaluated our models using FID metric as detailed in Appendix C.3. Since there is a high correlation between both these metrics, we use CLIP-score as primary metric that captures the alignment between individual image-caption pairs, as compared to general statistics of image distribution captured by FID. We put the plots and tables corresponding to FID results below, please note that lower FID is better.

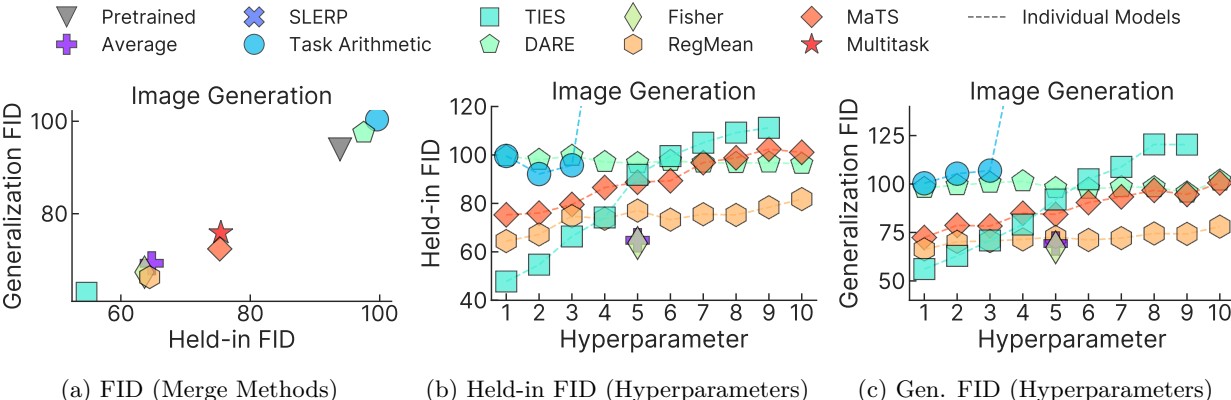

(a) FID (Merge Methods)       (b) Held-in FID (Hyperparameters)       (c) Gen. FID (Hyperparameters)

Figure 10: **FID of different merging methods across various hyperparameters in DomainNet generation.** These FID results complement the results provided in Fig. 2 and Fig. 4.

Table 24: **FID of different merging methods on the held-in tasks in the DomainNet generation setup for different hyperparameters.** See Appendix F for a descriptions of the hyperparameters. For methods without hyperparameters, we set the hyperparameter index to 5.

| Method | \multicolumn{10}{c}{Hyperparameter Index} |
|---|---|---|---|---|---|---|---|---|---|---|
| | 1 | 2 | 3 | 4 | 5 | 6 | 7 | 8 | 9 | 10 |
| Pretrained | | | | | 93.8 | | | | | |
| Average | | | | | 64.7 | | | | | |
| Task Arith. | 99.5 | 92.1 | 95.7 | 182.7 | 323.8 | 263.6 | 574.5 | 569.1 | 545.4 | 612.5 |
| TIES | 47.7 | 54.7 | 66.3 | 74.2 | 91.7 | 99.5 | 105.0 | 109.2 | 111.2 | |
| DARE | 99.5 | 97.9 | 99.6 | 97.0 | 96.6 | 97.5 | 96.8 | 96.6 | 96.8 | 96.3 |
| Fisher | | | | | 32.9 | | | | | |
| RegMean | 64.4 | 67.1 | 74.7 | 73.6 | 77.0 | 73.2 | 75.5 | 75.2 | 78.4 | 81.7 |
| MaTS | 75.2 | 76.0 | 79.5 | 86.5 | 88.7 | 89.3 | 96.8 | 98.8 | 102.4 | 101.0 |
| Multitask | | | | | 75.4 | | | | | |

## K   Qualitative Results of Image Generation

We provide qualitative samples generated by our merged models from the experiments in Appendix C.3. For this, we sample 6 unique captions from the held-in and generalization splits, and visualize generated images from the merged models below Fig. 11 and Fig. 12. Please find more samples in supplementary.

Table 25: **FID of different merging methods on the generalization tasks in the DomainNet generation setup for different hyperparameters.** See Appendix F for a descriptions of the hyperparameters. For methods without hyperparameters, we set the hyperparameter index to 5.

| Method | Hyperparameter Index | | | | | | | | | |
|---|---|---|---|---|---|---|---|---|---|---|
| | **1** | **2** | **3** | **4** | **5** | **6** | **7** | **8** | **9** | **10** |
| Pretrained | | | | | 93.9 | | | | | |
| Average | | | | | 69.2 | | | | | |
| Task Arith. | 100.3 | 105.3 | 106.9 | 188.9 | 321.5 | 292.6 | 603.6 | 606.7 | 580.2 | 653.7 |
| TIES | 56.1 | 62.9 | 70.9 | 79.0 | 92.0 | 102.4 | 108.8 | 120.4 | 120.3 | |
| DARE | 97.9 | 99.4 | 100.6 | 101.4 | 97.9 | 97.5 | 98.3 | 98.3 | 95.5 | 101.7 |
| Fisher | | | | | 67.3 | | | | | |
| RegMean | 66.2 | 70.2 | 70.6 | 71.4 | 72.6 | 71.1 | 72.0 | 74.4 | 74.2 | 78.0 |
| MaTS | 72.3 | 78.6 | 78.3 | 84.9 | 84.3 | 90.4 | 93.3 | 96.7 | 94.6 | 100.5 |
| Multitask | | | | | 75.8 | | | | | |

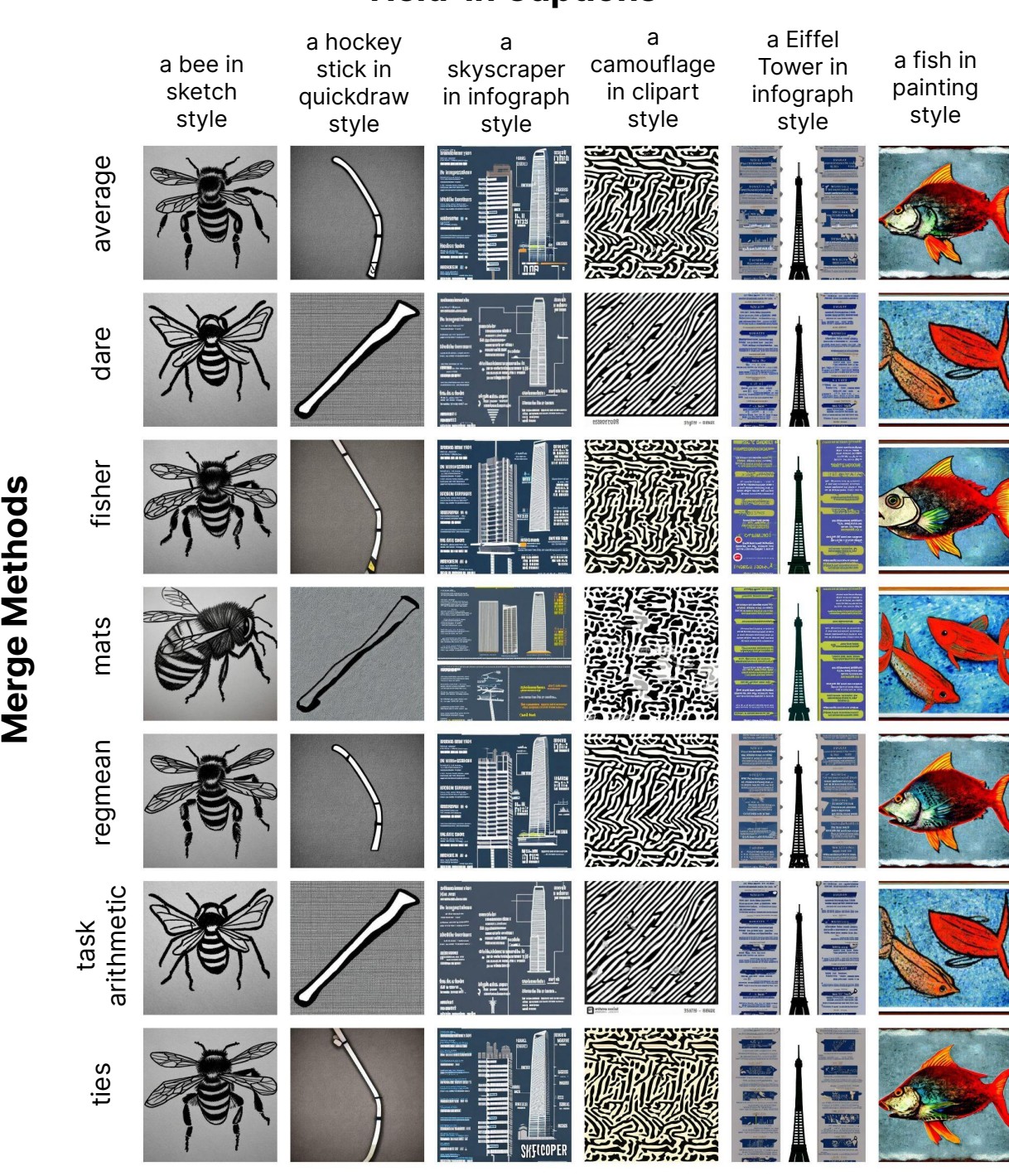

Figure 11: **Qualitatively comparing merging methods across captions in held-in set.** We use the best hyperparameters found by sweeping ranges mentioned in Appendix F.

# Generalization Captions

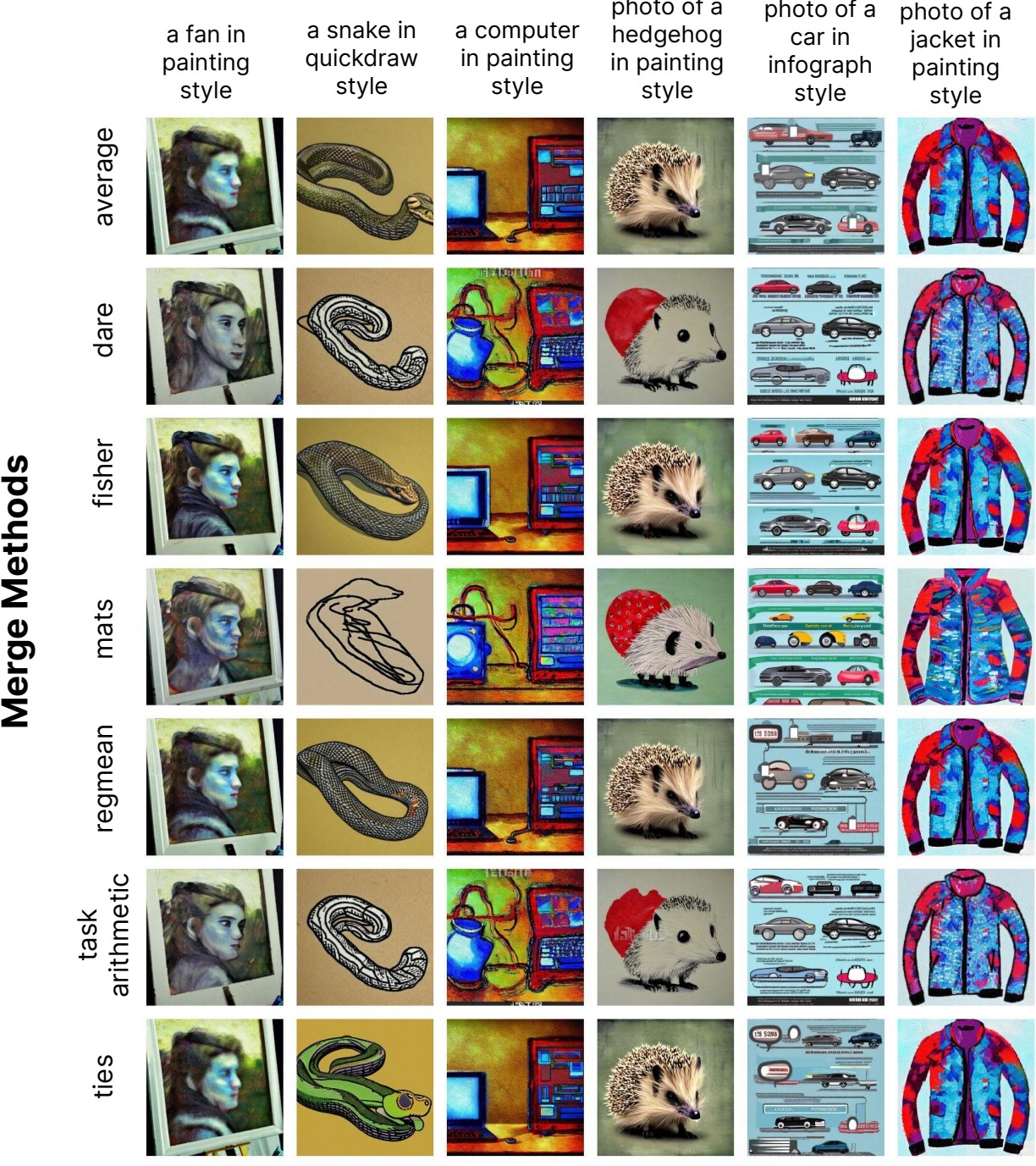

Figure 12: **Qualitatively comparing merging methods across captions in generalization set.** We use the best hyperparameters found by sweeping ranges mentioned in Appendix F.

