# OpenReview forum: "Realistic Evaluation of Model Merging for Compositional Generalization"
_TMLR — Accepted by TMLR_

### Review · Reviewer_rw6V · 2026-04-11

**Summary Of Contributions:**

- This paper presents a benchmark study that systematically compares model merging methods under a unified experimental framework.
- It goes beyond prior work that primarily evaluates held-in performance by introducing compositional generalization as a new evaluation criterion and analyzing it empirically.
- The authors conduct extensive experiments across three modalities (e.g.image classification, image generation, and natural language processing).
- In addition to performance, the paper considers practical aspects such as computational cost, prerequisites, hyperparameter sensitivity, and scaling behavior with respect to the number of merged models.

## Strengths ##
- Clearly identifies the lack of standardization in prior work and provides a structured attempt to address it.
- Introduces compositional generalization as a meaningful and complementary evaluation criteria.
- Conducts a comprehensive empirical study across multiple modalities and settings

## Weaknesses ##
- The paper lacks algorithmic novelty and is primarily an analysis-driven study.
- The definition and setup of compositional generalization may depend on specific scenarios, with limited discussion on its general applicability.
- Some experimental settings (e.g., cross-lingual NLP) are inherently challenging, which may overemphasize the limitations of merging methods.

**Audience:**

Yes

**Audience Explanation:**

- Model merging is widely used in practice, particularly in the context of LLMs and diffusion models, making the topic relevant to both researchers and practitioners.
- As a benchmark study, this paper provides a structured comparison that can help guide future research directions. Insights on compositional generalization, scaling behavior, and compute trade-offs are valuable for broader areas such as multi-task learning and model composition.
- The paper has the lack of methodological novelty which may limit its appeal to readers primarily interested in new algorithmic contributions.

**Broader Impact Concerns:**

This work primarily analyzes model merging methods and does not introduce direct ethical risks.

**Claims And Evidence:**

Yes

**Claims Explanation:**

- The paper provides sufficient empirical evidence by applying a consistent experimental protocol across multiple modalities (vision, generation, NLP).
- The study convincingly demonstrates previously overlooked issues (e.g., lack of comparability and practical considerations) through systematic experiments.
- However, some claims (e.g., the central importance of compositional generalization) are supported empirically but would benefit from a deeper discussion of their conceptual validity.

**Requested Changes:**

- Provide a clearer justification that the definition and evaluation of compositional generalization are fair and consistent **across all modalities**.
- Better **disentangle the effects of task difficulty** (especially in cross-lingual NLP) from the limitations of merging methods when interpreting results.
- Clarify the**selection criteria for merging methods** and discuss whether the chosen set is sufficiently representative of **recent advances**.

- Provide more concrete practical guidelines on when to use each merging method in real-world scenarios.
- Include additional ablation studies or robustness analyses for key experimental settings (e.g., DomainNet).
- Add case studies or qualitative examples demonstrating how compositional generalization translates to real-world applications.

---

> ### Author Response · Authors · 2026-05-14
> **Response to Reviewer rw6v**
>
> Thank you for your constructive feedback to help improve the paper.
>
> >The paper lacks algorithmic novelty and is primarily an analysis-driven study.
>
> The contribution of our work is to compare various merging methods in a realistic setup, focusing on compositional generalization, while considering other factors such as compute cost and hyperparameter sensitivity so that practitioners can know which method to use.
>
> >The paper has the lack of methodological novelty which may limit its appeal to readers primarily interested in new algorithmic contributions.
>
> Though our paper may not appeal to readers primarily interested in new algorithmic contributions, it can appeal to readers primarily interested in knowing which merging method to use in practice, when accounting for various factors.
>
> >The definition and setup of compositional generalization may depend on specific scenarios, with limited discussion on its general applicability.
>
> We have added a paragraph to section 3 defining and justifying compositional generalization in the different modalities.
>
> Concretely, consider a model fine-tuned on a task requiring skills A and B, and another model fine-tuned on a task requiring skills A' and B'. Compositional generalization measures whether a model can solve a task requiring skills A and B' or a task requiring skills A' and B. In the vision setting, one skill corresponds to operating across domains or styles (e.g., clipart vs. real images), while the other corresponds to image classification or generation (e.g. such as identifying the type of fruit). In the cross-lingual NLP setting, one skill corresponds to generating text in a particular language (e.g., English or Arabic), while the other corresponds to the task (e.g., summarization or question answering). We intentionally consider skills that are approximately orthogonal, meaning that learning one skill is not expected to directly improve performance on the other. For example, learning Korean should not inherently improve summarization ability, and learning to process clipart-style images should not improve fruit classification accuracy. By measuring compositionality of skills that are roughly "orthogonal", this ensures that any gain in performance comes from actually "composing" the "orthogonal" skills rather than a transfer of skills.
>
> > Some experimental settings (e.g., cross-lingual NLP) are inherently challenging, which may overemphasize the limitations of merging methods.
>
> We have updated the intro to emphasize the difficulty of our NLP setup involving generalizing across languages and the success of merging for image generation.
>
> >However, some claims (e.g., the central importance of compositional generalization) are supported empirically but would benefit from a deeper discussion of their conceptual validity.
>
> Please see response above.
>
> >Provide a clearer justification that the definition and evaluation of compositional generalization are fair and consistent across all modalities.
>
> Please see response above.
>
> >Better disentangle the effects of task difficulty (especially in cross-lingual NLP) from the limitations of merging methods when interpreting results.
>
> NLP tasks such as summarization are more difficult than vision tasks such as classifying the type of fruit in an image. Because the NLP tasks are more difficult, cross-lingual generalization for NLP could be more difficult. We have updated section 4 with this.
>
> >Clarify theselection criteria for merging methods and discuss whether the chosen set is sufficiently representative of recent advances.
>
> We wanted a representative set of methods that covered different approaches to merging.
>
> >Provide more concrete practical guidelines on when to use each merging method in real-world scenarios.
>
> Overall, we find that TIES generally provides a good trade-off between performance and various practical considerations such as prerequisites, compute, and hyperparameter tuning. We have added this to the introduction and conclusion.
>
> >Include additional ablation studies or robustness analyses for key experimental settings (e.g., DomainNet).
>
> In the DomainNet setting for scaling up the number of models, to account for variation in the models sampled to merge, we sample the models to merge 20 times. For each sample, we first sample the models to merge, then do a hyperparameter sweep for each merge, and then finally merge the models. This helps ensure the conclusions we draw, especially in the setup where we scale up the number of models to merge, is robust.
>
> >Add case studies or qualitative examples demonstrating how compositional generalization translates to real-world applications.
>
> We have added real-world examples with merging instruction-following models and merging robotics models to the introduction.

---

### Review · Reviewer_FJit · 2026-05-13

**Summary Of Contributions:**

This paper presents a unified evaluation of eight model merging methods across image classification, image generation, and cross-lingual NLP. It focuses on compositional generalization, not only held-in multitask performance, and analyzes practical factors such as prerequisites, compute, hyperparameter sensitivity, and scaling with the number of merged models. The main finding is that no method dominates across settings, and held-in performance can correlate with generalization in vision but anticorrelate in NLP.

**Audience:**

Yes

**Audience Explanation:**

The paper is relevant to readers interested in model merging, model reuse, multitask learning, and compositional generalization. Its main value is as an empirical benchmark and practical guide rather than a method paper.

**Claims And Evidence:**

Yes

**Claims Explanation:**

The empirical study is useful and mostly supports the paper’s main qualitative claims.

However, the “compositional generalization” framing needs more caution. In DomainNet, category-domain recombination may also reflect domain shift rather than clean skill composition. In cross-lingual NLP, failures may reflect catastrophic forgetting or multilingual transfer issues, not only limitations of merging.

The paper should also clarify hyperparameter selection. Since several methods are sensitive to hyperparameters, it is important to know whether selection uses held-in validation data, generalization data, or oracle test performance. The main results should also include uncertainty estimates, especially where differences are small.

**Requested Changes:**

Clarify the hyperparameter selection protocol, especially whether generalization-task data is used.

Add confidence intervals, standard deviations, or significance tests for the main results.

Include constituent-model performance on generalization tasks as a baseline.

Better justify or soften the compositional generalization interpretation.

Fix the inconsistency in the scaling experiment: the main text mentions 10 samples, Appendix G mentions 20, and the NLP tables mention 5.

---

> ### Author Response · Authors · 2026-05-14
> **Response to Reviewer FJit**
>
> Thank you for your thorough review and detailed suggestions to help improve the paper.
>
> >However, the “compositional generalization” framing needs more caution. In DomainNet, category-domain recombination may also reflect domain shift rather than clean skill composition. In cross-lingual NLP, failures may reflect catastrophic forgetting or multilingual transfer issues, not only limitations of merging.
>
> Learning across different domains or different languages can also be viewed as a “skill”. We have updated the draft to make it clear that skills can include different domains or different languages in section 3. Also, we have added to the intro that merging cannot outperform multitask training for the very difficult task of cross lingual NLP in section 1. If there is some terminology that works better, we would be happy to change the term compositional generalization.
>
> >The paper should also clarify hyperparameter selection. Since several methods are sensitive to hyperparameters, it is important to know whether selection uses held-in validation data, generalization data, or oracle test performance. The main results should also include uncertainty estimates, especially where differences are small.
>
> We have added a sentence in section 5.3 clarifying that the hyperparameter selection is done using validation data on the held-in datasets only.
>
> >Clarify the hyperparameter selection protocol, especially whether generalization-task data is used.
>
> Please see response above.
>
> >Add confidence intervals, standard deviations, or significance tests for the main results.
>
> We no longer have access to the individual run performance for the scaling experiments, only the average scores, so we cannot compute confidence intervals/standard deviations.
>
> >Include constituent-model performance on generalization tasks as a baselines.
>
> For constituent-model performance, the models were individually trained on the held-in tasks, so we would not expect them to perform well at all on the tasks. Also, it is not clear which constituent model we should use when evaluating on the generalization tasks since none of them were trained on that task. We include the pre-trained model as a baseline, which we expect to do similar on the generalization task as the constituent models that were trained on other tasks.
>
> >Better justify or soften the compositional generalization interpretation.
>
> We have better justified the compositional generalization interpretation
> Concretely, consider a model fine-tuned on a task requiring skills A and B, and another model fine-tuned on a task requiring skills A' and B'. Compositional generalization measures whether a model can solve a task requiring skills A and B' or a task requiring skills A' and B. In the vision setting, one skill corresponds to operating across domains or styles (e.g., clipart vs. real images), while the other corresponds to image classification or generation (e.g. such as identifying the type of fruit). In the cross-lingual NLP setting, one skill corresponds to generating text in a particular language (e.g., English or Arabic), while the other corresponds to the task (e.g., summarization or question answering). We intentionally consider skills that are approximately orthogonal, meaning that learning one skill is not expected to directly improve performance on the other. For example, learning Korean should not inherently improve summarization ability, and learning to process clipart-style images should not improve fruit classification accuracy. By measuring compositionality of skills that are roughly "orthogonal", this ensures that any gain in performance comes from actually "composing" the "orthogonal" skills rather than a transfer of skills.
>
> >Fix the inconsistency in the scaling experiment: the main text mentions 10 samples, Appendix G mentions 20, and the NLP tables mention 5.
>
> We have updated the text to make it consistent and clarify this. The vision experiments use 20 samples and the NLP experiments use 5 samples.

---

### Review · Reviewer_LEFN · 2026-05-13

**Summary Of Contributions:**

### Overall
The paper evaluates a set of methods of model merging for compositional generalization. The authors claim to address the lack of standardization in how merging techniques are compared. Comprehensive experiment results are included in the paper.

### Strengths
- The paper covers eight merging methods across three distinct modalities (image classification, image generation, cross-lingual NLP). This is much broader than prior single-modality benchmarks and makes the conclusions more credible.
- Reframing the evaluation away from held-in multitask performance, which the authors correctly note can be trivially achieved by just using the constituent model, toward compositional generalization captures a more realistic and useful application of merging.
- The anticorrelation between held-in and generalization performance in NLP, and the cross-modality variation in best-performing method, are quite informative results that justify the unified-evaluation framing.

### Weaknesses
- Several relevant recent methods (Tangent Task Arithmetic, AdaMerging, Concrete Subspace, evolutionary merging, weight-ensembling MoE, ZipIt, Git Re-Basin) are acknowledged in Section 7 but excluded.
- The setup essentially tests cross-domain transfer (classify fruit in clipart given fruit-in-real and tool-in-clipart) and cross-lingual transfer. For instance, the image-generation evaluation conflates style transfer with compositionality. Does that really represent compositional generalization?

**Audience:**

Yes

**Audience Explanation:**

The authors provide a unified evaluation for model merging, which will be interesting for many individuals in TMLR's audience.

**Broader Impact Concerns:**

I have no broader impact concerns.

**Claims And Evidence:**

Yes

**Claims Explanation:**

The paper's broader claims about the need for unified evaluation and the existence of cross-modality variation are well-supported.

**Requested Changes:**

There are several requested changes:
- Add the experiments for missing methods.
- Justify the "compositional generalization" terminology as stated in weaknesses.
- From Fig.2 (right), it seems Fisher Merging in particular appears to drive the NLP anticorrelation. I wonder if the conclusion will be the same if excluding Fisher Merging (some sensitivity analysis).

---

> ### Author Response · Authors · 2026-05-14
> **Response to Reviewer LEFN**
>
> Thank you for your helpful feedback for improving the paper.
>
> >Several relevant recent methods (Tangent Task Arithmetic, AdaMerging, Concrete Subspace, evolutionary merging, weight-ensembling MoE, ZipIt, Git Re-Basin) are acknowledged in Section 7 but excluded.
>
> We focused on more popular merging methods. Tangent task arithmetic does not merge the original fine-tuned models, but requires fine-tuning modes in a different setup using linearized fine-tuning. AdaMerging requires loading all the models into memory and leaving them in memory to compute the gradient, which is not possible with the hardware used to run the experiments. ZipIt and Git Re-Basin are merging methods intended to handle models with different architectures and would just default to simple averaging over models with the same architecture, which is the setup we have. Weight-ensembling MoE results in a final model that is an MoE that requires more memory to load.
>
> >The setup essentially tests cross-domain transfer (classify fruit in clipart given fruit-in-real and tool-in-clipart) and cross-lingual transfer. For instance, the image-generation evaluation conflates style transfer with compositionality. Does that really represent compositional generalization?
>
> Learning across different skills can also be viewed as a “skill” so we view style transfer across different classification tasks as compositionality. We have updated the draft to make it clear that skills can include different domains or different languages in section 3. Also, we have added to the intro that merging cannot outperform multitask training for the very difficult task of cross lingual NLP in section 1. If there is some terminology that works better, we would be happy to change the term compositional generalization.
>
> >Add the experiments for missing methods.
>
> Please see response above.
>
> >Justify the "compositional generalization" terminology as stated in weaknesses.
>
> Please see response above.
>
> >From Fig.2 (right), it seems Fisher Merging in particular appears to drive the NLP anticorrelation. I wonder if the conclusion will be the same if excluding Fisher Merging (some sensitivity analysis).
>
> We checked how the correlation changed when excluding Fisher merging. Excluding Fisher merging, the magnitude of the correlation decreases from -.853 to -.674.

---

### Decision · Action_Editor_wYYt · 2026-06-21

**Recommendation:** Accept with minor revision

**Additional Comments:**

Some remaining concerns should be addressed or discussed in the final version. For example, as Reviewer rw6V mentioned, the paper is primarily an empirical benchmark rather than a methodologically novel contribution, and the cross-lingual NLP setting still makes it somewhat difficult to fully disentangle task difficulty from the intrinsic limitations of merging methods. The selection criteria for the included merging methods could also be stated more explicitly.

**Audience:**

Yes

**Audience Explanation:**

This paper focuses on model merging for multiple pretrained models. This is a popular topic that many machine learning researchers are interested in.

**Claims And Evidence:**

Yes

**Claims Explanation:**

This paper presents a unified evaluation of eight model merging methods across image classification, image generation, and cross-lingual NLP. Extensive experimental results are provided.

---

> ### Author Response · Authors · 2026-07-14
> **Response**
>
> Thank you for reviewing our work and the helpful feedback. We have made the following update to the new camera-ready version.
>
> 1. Added in section 4
> Because cross-lingual generalization is a very difficult task, it is not clear if merging methods underperform multitask training due to the difficulty of cross-lingual generalization or due to some limitations of current merging methods.
>
> 2. Added in section 2
> An exhaustive comparison of merging methods is beyond the scope of this work, so we choose ones that represent a diversity of approaches (i.e. that minimize interference between models or minimize the distance in activations between the models) and that represent more widely used methods and less common methods.
>
> 3. Added at end of intro
> As a whole, this paper presents a benchmark for comparing various merging methods for both held-in tasks and compositional generalization across various modalities.